

# Evapotranspiration monitoring based on thermal infrared data over agricultural landscapes: comparison of a simple energy budget model and a SVAT model

Guillaume Bigeard[1,2], Benoit Coudert[1,2], Jonas Chirouze[1], Salah Er-Raki[3], Gilles Boulet[1,4],
Eric Ceschia[1,2], and Lionel Jarlan[1,3]

[1]CESBIO, Toulouse, France
[2]Université Paul Sabatier, Toulouse, France
[3]Université Cadi Ayyad, Marrakech, Morocco
[4]IRD, Institut National Agronomique de Tunisie, Tunis, Tunisia

**Correspondence:** B. Coudert (benoit.coudert@cesbio.cnes.fr)

**Abstract.** The overall purpose of our work is to take advantage of Thermal Infra-Red (TIR) imagery to estimate landscape evapotranspiration fluxes over agricultural areas, relying on two approaches of increasing complexity and input data needs : a Surface Energy Balance (SEB) model, TSEB, used directly at the landscape scale with TIR forcing, and the aggregation of a Soil-Vegetation-Atmosphere Transfer (SVAT) model, SEtHyS, run at high resolution ($\simeq 100$ m) and constrained by assimilation

of TIR data. Within this preliminary study, models skills are compared thanks to large *in situ* database covering different crops, stress and climate conditions. Domains of validity are assessed and the possible loss of performance resulting from inaccurate but realistic inputs (forcing and model parameters) due to scaling effects are quantified. The *in situ* data set came from 3 experiments carried out in southern France and in Morocco. On average, models provide half-hourly averaged estimations of latent heat flux ($LE$) with a RMSE of around 55 $Wm^{-2}$ for TSEB and 47 $Wm^{-2}$ for SEtHyS, and estimations of sensible heat

flux ($H$) with a RMSE of around 29 $Wm^{-2}$ for TSEB and 38 $Wm^{-2}$ for SEtHyS. TSEB has been shown to be more flexible and requires one single set of parameters but lead to low performances on rising vegetation and stressed conditions. An in-depth study on the Priestley-Taylor key parameter highlights its marked diurnal cycle and the need to adjust its value to improve flux partition between sensible and latent heat fluxes (1.5 and 1.25 for south-western France and Morocco, respectively). Optimal values of 1.8 to 2 were hilighted under cloudy conditions, which is of particular interest with the emergence of low

altitude drone acquisition. SEtHyS is valid in more cases while it required a finer parameters tuning and a better knowledge of surface and vegetation. This study participates to lay the ground for exploring the complementarities between instantaneous and continuous dynamic evapotranspiration mapping monitored with TIR data.

## 1 Introduction

Exchange of water at the soil-vegetation-atmosphere interface is of prime importance for weather forecasting and for climate

studies (Shukla and Mintz, 1982); it is also a key component for hydrology, and therefore catchment water balance (Milly, 1994), and for agronomy in order to improve irrigation scheduling (Allen et al., 1998). Despite the abundant literature on the





subject, there is no consensual approach for its spatialized estimation, and the contribution of evapotranspiration (ET) to the global hydrologic cycle remains uncertain (Jasechko et al., 2013). There are several *in situ* techniques available to measure ET (Allen et al., 2011) but most suffer from a lack of spatial representativeness. This prevents their use as a sustainable solution for regional applications, especially for agricultural landscape where spatial heterogeneity –in terms of farming and technical itineraries including the resulting pattern of moisture conditions– is high. By contrast, remote sensing offers an attractive alternative through the synoptic and repeated data acquisition it provides. Indeed, even if ET is not directly observable from space, remote sensing data in different parts of the electromagnetic spectrum are related to the characteristics of the land surface governing the evapotranspiration process.

Within this context, several approaches combining remote sensing data and land surface models of various complexity were proposed for the regional monitoring of ET (Courault et al., 2005), from the most conceptual approaches modulating the evaporative demand by an empirical coefficient (the so called "crop coefficient", Allen et al., 1998), to the complex and mechanistically-based Soil-Vegetation-Atmosphere Transfer (SVAT) models that require a large number of inputs. In-between, the surface energy balance (SEB) models, constrained by thermal-infrared radiative temperature observations, have been gaining influence over the last decade (Choi et al., 2009; Chirouze et al., 2014; Diarra et al., 2017). Several authors intercompared the different SEB-based approaches for mapping ET with noticeable discrepancies (see Zhan et al., 1996; French et al., 2005; Timmermans et al., 2007, 2011). By contrast, Kalma et al. (2008) reviewed the different methods to estimate surface evapotranspiration based on surface temperature derived from remote sensing data and highlights similar accuracies on average. Some other studies (Franks and Beven, 1999; Schuurmans et al., 2003; Meijerink et al., 2005; Crow et al., 2005) discussed the complementarities between simple instantaneous SEB and more complex SVAT modeling approaches to improve surface turbulent fluxes estimations through data assimilation thanks to their independent modeling errors. Nevertheless, both methodologies, SEB or SVAT-based, were rarely compared in literature (apart from Crow et al., 2005) due to the different underlying diagnostic or prognostic equations of the models with respect to the distinct purposes of the approaches in terms of temporal and/or spatial resolutions of evapotranspiration estimates. Either based on SVAT or SEB models, the estimation of surface evapotranspiration implies dealing with the method-model complexity issue (Carlson, 2007; Kalma et al., 2008), and with the always incomplete knowledge to document or to constrain them. This has partly to do with the trade-off between revisit frequency and resolution for actual TIR sensors. For the higher resolution, another source of uncertainty is coming from to the surface temperature fluctuations in relation with atmospheric turbulence (Lagouarde et al., 2013). Remotely sensed TIR data are available either at high spatial resolution and low temporal frequency or vice versa. McCabe and Wood (2006) have shown how spatial resolution of TIR data used as input in SEB method impacted the spatial variation of flux estimates. The lack of knowledge on scaling effects when fluxes are intercompared at the same scales using aggregation or disaggregation methods was pointed out by several authors (Kustas et al., 2003, 2004; Norman et al., 2003). Within this context, the overall purpose of our work is to compare both methodologies in order to estimate spatially distributed evapotranspiration on agricultural landscapes at high ($\simeq$100m) and intermediate ($\simeq$1km) spatial resolutions. This study specifically focuses on the comparison of the TSEB model (Norman et al., 1995) and the SEtHyS SVAT model (described in Coudert et al., 2006) constrained with TIR data. The more specific purpose is twofold: (1) accuracy assessment through the confrontation of both methodologies to a large *in*




*situ* database acquired in the South West of France and in Morocco with a discussion on the domain of validity of the models; (2) a comprehensive sensitivity analysis to uncertainties in terms of inputs and parameters and a discussion on the selection strategy of the parameters set. This paper is organized as follows. After briefly introducing data sets and both models (Sect. 2), the analysis of the models performances is presented (Sect. 3.1). Then, we focus on sensitivity analysis results (Sect. 3.2) and

on discussions related to parameters and inputs (Sect. 4). Finally, conclusions and perspectives are drawn in Sect. 5.

## 2 Data and methods

### 2.1 Models description

The formulation of the two-source energy balance system, which is similar in both models is firstly described. Then, differences in the solving method and associated assumptions, together with differences in flux parameterization, are briefly reminded.

#### 2.1.1 The two-source energy budget

In the two-source energy balance, total sensible ($H$) and total latent heat ($LE$) fluxes arise from the soil and vegetation heat and vapor sources. Applying energy conservation and continuity principles, the energy budget can be described with the following set of equations:

$$H = H_{[soil]} + H_{[veg]} \tag{1}$$

$$LE = LE_{[soil]} + LE_{[veg]} \tag{2}$$

$$R_n = R_{n[soil]} + R_{n[veg]} \tag{3}$$

$$R_{n[soil]} = H_{[soil]} + LE_{[soil]} + G \tag{4}$$

$$R_n = H + LE + G, \tag{5}$$

where $G$ is the ground heat flux and $R_n$ is the net radiation. All fluxes are expressed in $Wm^{-2}$. The $H$ and $LE$ fluxes

expressions are given in Shuttleworth and Wallace (1985, Eq. 6 and 7, p. 843) for a resistive scheme (following analogy with Ohm's law) of a one-dimensional description of energy partition for sparse crops assuming horizontal uniformity. $H$ and $LE$ expressions for the complete canopy between the level of mean canopy fow and reference height can then be written as:

$$H = -\frac{\rho c_p}{r_a^a}(T_x - T_0) \tag{6}$$

$$LE = -\frac{\rho c_p}{r_a^a \gamma}(e_x - e_0) \tag{7}$$

Where $\gamma$ is the psychrometric constant ($mb\ K^{-1}$), $r_a^a$ the aerodynamic resistance between canopy source height and reference level ($s\ m^{-1}$), $e_x$ and $e_0$ vapor pressure ($mb$) at canopy source height and reference height and $T_x$ and $T_0$ temperature





($C$) at canopy source height and at reference height. The components elements from soil and vegetation ($LE_{[soil]}$, $LE_{[veg]}$, $H_{[soil]}$, and $H_{[veg]}$) are expressed in the same way according to the associated resistances. Afterwards, the vapor pressure deficit at the canopy source height is introduced. The system now becomes a set of five equations with six unknowns, namely: vegetation temperature $T_{[veg]}$, soil temperature $T_{[soil]}$, canopy-space temperature $T_{[canopy]}$ and the corresponding water vapour

pressure $e_{[veg]}$, $e_{[soil]}$ and $e_{[canopy]}$. The next steps of the classical solving of a two-source energy balance system are to express $T_{[canopy]}$ as a function of $T_{[veg]}$ and $T_{[soil]}$ thanks to the continuity equation in $H$ and $T_{[veg]}$ as a function of $T_{[soil]}$ using the energy budget of vegetation. In addition, the heat conduction flux in soil $G$ is either estimated from net radiation (TSEB model) or residual of the energy budget (SEtHyS model) as detailed in the appendix. The solving method consists in the linearization of the equations of the previous system. The basic differences between approaches is that for SVATs models, soil temperatures

at different depths are prognostic variables tightly linked to water mass balance, whilst radiative temperature is a forcing input for the SEB models used to infer $T_{[veg]}$ and $T_{[soil]}$ as detailed below.

### 2.1.2 TSEB

The TSEB model is fully described in Norman et al. (1995). The solving principle is briefly described below. The TSEB model relies on several assumptions and approximation to bypass the evaluation of the $T_{[soil]}$ prognostic variable. First, it is forced by

a radiometric surface temperature $T_{rad}$ so that soil and vegetation temperatures contribute to $T_{rad}$ in proportion to the fraction of the radiometer field of view ($f_\theta$) that is occupied by each component, thus adding a sixth equation to the system above:

$$T_{rad}(\theta) = [f_\theta \times T_{[veg]}^n + (1 - f_\theta) \times T_{[soil]}^n]^{1/n}, \tag{8}$$

where the factor n is usually fixed to 4 (Becker and Li, 1990). The available energy at the soil surface is computed considering an exponential extinction of net radiation (i.e. Beer's Law):

$$R_{n[soil]} = R_n \times e^{-\kappa \times LAI}, \tag{9}$$

where the factor $\kappa$ is set to 0.45 for spherical distribution of leaves following Roos (1991). The conduction flux in the soil is expressed as a fraction of the available energy at the soil surface :

$$G = \Gamma \times R_{n[soil]}, \tag{10}$$

with $\Gamma$ an empirical coefficient taken as 0.35 (Choudhury et al., 1987). Finally, the resolution of this set of equations relies

on the (strong) assumption that, most of the time, vegetation transpires at a potential rate. The Priestley Taylor equation gives a first estimation of canopy transpiration (Norman et al., 1995, Eq. 12):

$$LE_{[veg]} = \alpha_{PT} \times f_g \times \frac{\Delta}{\Delta + \gamma} \times R_{n[veg]}, \tag{11}$$





where $\alpha_{PT}$ is the Priestley Taylor parameter, $f_g$ the green vegetation fraction cover, $\Delta$ the slope of the saturation vapor pressure versus temperature curve and $\gamma$ the psychrometric constant.

In the "series" resistance network used in this study (see justification below) described in Norman et al. (1995, Fig. 11), the sensitive heat fluxes are expressed as:

$$H_{[soil]} = \rho c_p \frac{T_{[soil]} - T_{[canopy]}}{r_s} \tag{12}$$

between the soil surface and the canopy air space,

$$H_{[veg]} = \rho c_p \frac{T_{[veg]} - T_{[canopy]}}{r_x} \tag{13}$$

between vegetation and canopy air space,

$$H = \rho c_p \frac{T_{[canopy]} - T_a}{r_a} \tag{14}$$

between canopy air space and reference height for atmospherical measurements. Where $r_s$, $r_x$ and $r_a$ are the associated resistances given respectively in (Norman et al., 1995, Eq. B.1, Eq. A.8, Eq. 6).

$H_{[veg]}$ is then computed as the residual of the vegetation energy balance (eq. 1). $T_{[veg]}$ is derived from $H_{[veg]}$; $T_{[soil]}$ from Eq. (8); $H_{[soil]}$ is computed from $T_{[soil]}$ and $LE_{[soil]}$ as a residual of the soil energy balance (Eq. 1). Should $LE_{[soil]}$ be found negative, meaning that there is condensation on the soil surface, which is very unlikely during the day, then the assumption of a vegetation transpiring at the potential rate is bypassed. $LE_{soil}$ is set to zero and a new $LE_{[veg]}$ value is computed. Likewise, should $LE_{[veg]}$ be found negative, $LE$ is set to zero. Further details on the resolution are given in French (2001, see his Fig. 2.6 p. 39). Compared to the initial formulation of the TSEB model (Norman et al., 1995), a more physically-based parameterization for the divergence of $R_n$ described in Kustas and Norman (1999) was adopted. In agreement with (Li and Kustas, 2005), the "series" layout of resistance (Norman et al., 1995) was found to provide overall more accurate results (not shown) and also less sensitivity to parameters uncertainty in the case of sparse canopy. Moreover, (Li and Kustas, 2005) have shown that the "parallel" resistance network was found to be more sensitive to errors in vegetation cover estimate. Finally, for model comparison, it was also relevant that both resistance network were similar in TSEB and SEtHyS model. For these different reasons, the "series" version of TSEB model was preferred to the "parallel" version in this study. The parameter sets used for TSEB are summarized in Table 1.

### 2.1.3 SEtHyS

The SEtHyS –French acronym for *Suivi de l'Etat Hydrique des Sols* or monitoring of the hydric condition of the soils– SVAT model physics and the main parameterizations are described in Coudert et al. (2006). The main equations of SEtHyS are summarized in appendix A. The model belongs to the "two sources, two layers" SVAT model category. Actually, the coupled



**Table 1.** TSEB parameters (9) with reference values and optimal values obtained from sensitivity analyses.

| Category | Parameter | Description [unit] | Litterature range | Reference value | Optimal value |
|----------|-----------|--------------------|--------------------|-----------------|---------------|
| **Optical properties** | $A_{soil}$ | Soil albedo | $0.05 - 0.35$ | 0.15 | 0.14 |
| | $A_{vegetation}$ | Vegetation albedo | $0.10 - 0.30$ | 0.3 | 0.3 |
| | $E_{soil}$ | Soil emissivity | $0.94 - 0.97$ | - | 0.94 |
| | $E_{vegetation}$ | Vegetation emissivity | $0.90 - 0.99$ | - | 0.97 |
| | $\epsilon$ | Surface emissivity, involved in CNR1 Ts conversion | $0.96 - 0.99$ | - | 0.96 |
| **Vegetation characteristics** | $S$ | Leaf size $[m]$, involved in computing surface resistance | - | 0.01 | 0.01 |
| | $\alpha_{PT}$ | Priestley-Taylor coefficient, involved in estimating canopy transpiration (Eq. 11) | $1 - 2$ | 1.26 | $1.3 - 1.5$ |
| **Surface properties** | $\Gamma$ | Soil energy partition coefficient : $G = \Gamma \times R_{n[soil]}$ (Eq. 10) | - | 0.35 | 0.35 |
| | $\kappa$ | Coefficient of the exponential extinction of net radiation to compute available energy at the soil surface (Eq. 9) | $0.3 - 0.6$ | 0.45 | 0.4 |

water and energy budget is solved for the vegetation and soil sources and the soil description for water and heat transfers is based on the force-restore Deardorff formalism (Deardorff, 1978). The model requires atmospheric and radiative forcing and surface biophysical parameters as inputs. It calculates the energy and water fluxes between surface and atmosphere and simulates the evolution of soil and canopy temperatures, air temperature and specific humidity within the canopy, as well as

the surface and the root zone soil water content. The heat and water transfer calculation within the continuum soil-vegetation-atmosphere is based on a resistance concept. The resistance network is made of four nodes: the reference height for the low atmospheric weather forcing; inside the vegetation at the displacement height plus the roughness length; just above ground at the soil roughness length; and, at ground level. The aerodynamic resistances –above and inside vegetation canopy– are determined with the wind speed profile description inside the canopy from Shuttleworth and Wallace (1985) and Lafleur and

Rouse (1990). The evapotranspiration calculation takes into account partition between free water in the canopy and the rest of the leaves (Monteith, 1965; Deardorff, 1978) and is based on the stomatal resistance for "big leaf" model from Collatz et al. (1991). The vegetation photosynthesis and stomatal resistance parameterizations are the same as those used by the SiB model (Sellers et al., 1996a). The soil hydrodynamic properties to calculate water transfer processes within the soil porous network are given by Genuchten (1980). Ground heat flux conduction is obtained as the residual of the soil energy budget. Finally, the

radiative transfer model included in the model for TIR domain (François, 2002) allows simulating brightness temperature and




**Table 2.** SetHyS parameters (22) with initial uncertainty ranges used for MCIP calibration.

| Category | Parameter | Description [unit] | Initial uncertainty range |
|---|---|---|---|
| **Optical properties** | $E_g$ | Bare soil emissivity | 0.94 – 0.99 |
| | $A_{sec}$ | Dry soil albedo | 0.225 – 0.35 |
| | $A_{hum}$ | Wet soil albedo | 0.1 – 0.22 |
| | $W_{inf}$ | Moisture parameter for albedo calculation | 0.15 – 0.29 |
| | $W_{sup}$ | Moisture parameter for albedo calculation | 0.291 – 0.5 |
| | $A_{sv}$ | Vegetation albedo | 0.16 – 0.32 |
| **Vegetation characteristics** | $V_{max0}$ | Leaf photosynthetic capacity (Rubisco) [$\mu mol\ m^{-2}\ s^{-1}$] | 30 – 200 |
| | $l_{gf}$ | Dimension of the leaf along the wind direction [$m$] | 0.01 – 0.08 |
| | $k_{wstr}$ | Empirical parameter for water stress calculation | 0.01 – 0.1 |
| **Ground properties** | $p_{hc}$ | "Half critic" hydrologic potential [$m$] | -200 – 100 |
| | $W_{max}$ | Saturated soil water content [$m^3\ m^{-3}$] | 0.3 – 0.5 |
| | $W_{resid}$ | Residual soil water content [$m^3\ m^{-3}$] | 0.05 – 0.15 |
| | $h_{VG}$ | Scale factor in the Van Genuchten retention curve model [$m$] | -1.161 – 0.251 |
| | $n_{VG}$ | Shape parameter in the Van Genuchten retention curve model | 1.168 – 1.331 |
| | $K_{sat}$ | Saturated hydraulic conductivity [$m\ s^{-1}$] | $2.4 \times 10^{-8} – 2.7 \times 10^{-6}$ |
| | $a_{Elim}$ | Empirical parameter for limit evaporation | 1 – 50 |
| | $b_{Elim}$ | Empirical parameter for limit evaporation | 1 – 50 |
| | $F_{therm}$ | Correction coefficient of the volumetric soil heat capacity [$J\ m^{-3}\ K^{-1}$] | 0.5 – 2 |
| | $dp_2$ | Root zone depth [$mm$] | 200 – 2000 |
| **Initialization variables** | $w_{g0}$ | Initial soil surface water content [$m^3\ m^{-3}$] | - |
| | $w_2$ | Initial root zone water content [$m^3\ m^{-3}$] | - |
| | $bias_{T2}$ | Error in deep soil temperature [$K$] | -2 – 2 |

radiative temperature, and thus gives the possibility of constraining the model with TIR data (Coudert and Ottlé, 2007; Coudert et al., 2008). The SEtHyS model requires a set of about 22 parameters presented in Table 2.





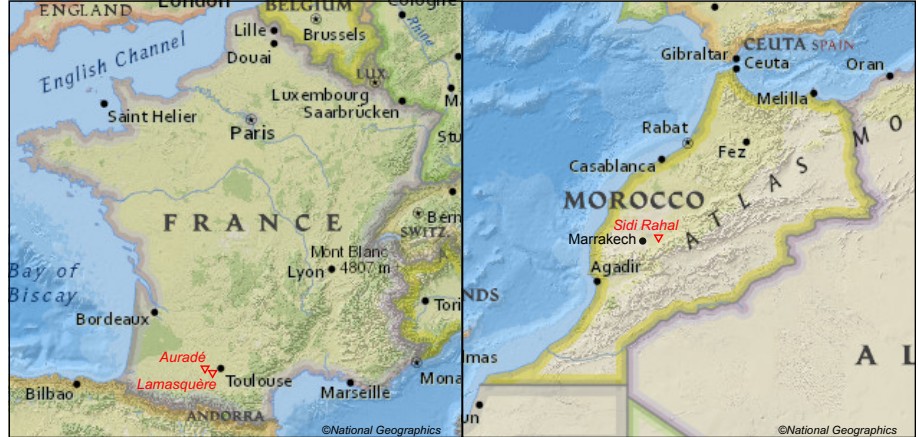

**Figure 1.** Experimental sites localization in France (left) and Morocco (right).

## 2.2 Sites description and data

All necessary data to run, calibrate and evaluate models was collected over 3 agricultural sites sampling 2 different climates (temperate and semi-arid) and 7 cultures cycles from seedling to harvest (wheat *Triticum aestivum L.*, sunflower *Helianthus annuus L.* and corn *Zea mays L.*), which enable us to characterize all phenological states and contrasted weather forcing.

Table 3 summarizes existing cultures and climates per site. Auradé (43.55°N, 1.11°E) and Lamasquère (43.50°N, 1.24°E) experimental sites are located near Toulouse in south-western France and are part of the "SudOuest" project (Dedieu et al., 2001; Béziat et al., 2009) led by CESBIO. Both experimental sites are under influence of temperate climate. They differ by management practices (culture rotation and irrigation), soil properties and topography. Complete description of site features and datasets are presented in Béziat et al. (2009). Sidi Rahal (31.67°N, 7.60°W) experimental site is located in the Haouz plain

in central Morocco and is part of the "Sud-Med" project (Chehbouni et al., 2008; Jarlan et al., 2015). It is part of an irrigated agricultural area under influence of a semi-arid climate. More information about site and dataset is given in Duchemin et al. (2006). Geographical localization of experimental sites is shown in fig. 1.

Each experimental station collected standard meteorological data at a half-hourly time step intervals: Global incoming shortwave and longwave radiation ($R_g$ and $R_a$), wind speed ($U_a$), air temperature ($T_a$), atmospheric pressure ($P_a$), relative

humidity ($R_h$) and rainfall. The four components of the net radiation ($R_n$) were measured using a CNR1 radiometer (Kipp and Zonen, Delft, NL). Land surface temperature (LST) was computed from measured upward and downward long wave components of the net radiation, using Stefan-Boltzmann's law and an estimation of surface emissivity (Becker and Li, 1995). Sensible ($H$) and latent ($LE$) heat fluxes were measured continuously using Eddy-Covariance (EC) systems (Moncrieff et al., 1997; Aubinet et al., 2000). Fluxes were processed with classical EC filters and corrections (Béziat et al., 2009). Accuracy on

flux estimates is expected to range between 5 % and 30 % (Eugster et al., 1997; Wilson et al., 2002). Soil heat flux ($G$) was sampled using heat flux plates located at depths ranging from 5 $cm$ to 1 $m$. Automatic measurements were then complemented





by vegetation sample. Vegetation height ($h_c$) and green Leaf Area Index ($LAI$) were collected periodically along crop cycles and interpolated using Piecewise cubic Hermite algorithm. Green $LAI$ was determined from destructive measurements with a LiCor planimeter (LI3100, LiCor, Lincoln, NE, USA). In order to obtain estimation of fraction of green ($f_g$), total $LAI$ ($LAI_{green} + LAI_{yellow}$) was extrapolated from green $LAI$ data, applying a linear decrease starting at the $LAI$ maximum and

ending at harvest with a value of $LAI_{total} = 0.8 \times LAI_{max}$. In order to assess the potential loss of accuracy of meteorological inputs at the landscape scale and impact on model simulations, SAFRAN reanalysis data (Quintana-Seguí et al., 2008) are used within this study. SAFRAN is based on an optimal interpolation between a background estimate obtained from Météo France's Numerical Weather Prediction Model (ALADIN) and weather station observations, except for precipitation relying on the ground station network only and for the incoming radiation fluxes (downwelling surface shortwave and longwave) which

are computed from Ritter and Geleyn's radiation scheme (1992) from the outputs of a numerical weather forecast model and the solar constant at the top of the atmosphere (for shortwave incoming radiation). Data was kindly provided by Météo-France.

## 2.3 Assessing the model skills

Specific periods of interests were identified to assess the model skills by phenological stages. These periods were chosen to be 10-day long in order to catch synoptic scale variability of the weather, as shown by Eugster et al. (1997) with the help of

spectral analysis. This duration is also short enough to remain representative of a specific phenological stage. Four specific studied periods spread along each crop cycles were chosen: rising phase, growth phase, maximum development phase and senescence. They were defined based on $LAI$ thresholds and variations throughout the growth cycle. Starting days were adjusted to optimize quantity and quality of data. In order to better assess the differences of model skills during stress periods, water stress is quantified using two indicators:

– the Evaporation Stress (SE, Boulet et al., 2007) related to the ratio between real and potential evapotranspirations:

$$SE = 1 - \frac{LE}{LE_{pot}}, \tag{15}$$

where $LE_{pot}$ was computed using the Penman-Monteith equation.

    – the Soil Wetness Index (SWI, Douville, 1998, among others) of the root zone ranging from 0 at wilting point to 1 at field capacity:

$$SWI = \frac{W_2 - W_{wilt}}{W_{fc} - W_{wilt}}, \tag{16}$$

with $W_2$ the root zone water content, $W_{fc}$ the water content at field capacity, and $W_{wilt}$ the water content at wilting point.

As cultures from our dataset are irrigated or in temperate areas, most stress periods are found during senescence phases, when water resources are low or irrigation is stopped. The assessment of model skills is assessed through classical statistical metrics

including Root Mean Square Error (RMSE), Mean Absolute Percentage Deviation (MAPD), bias and determination coefficient $r^2$.



### 2.4 Models calibration strategy

The calibration is based on the large experimental data set of both instrumented sites in South West of France (Béziat et al., 2009) and southeastern of Morocco (Chehbouni et al., 2008; Jarlan et al., 2015). The 22 parameters of the SEtHyS model were calibrated for each crops and each phenological stages. The objective was not clearly to calibrate the model to fit the data at

best but rather to evaluate the sensitivity of model outputs to potentially poorly calibrated parameters when a spatial application of the modeling tool is sought. Four different cases corresponding to four different set of parameters are considered to quantify the potential loss of performances due to wrong parameter values. The four cases are listed below from the "best" conditions when the parameters are calibrated for each site, each crop and each phenological stage to the worst when generic values are used:

1. Site and period specific parameters sets (hereafter named "optimal") for each site, crop class and phenological stages (i.e. the calibrated values of Sect. 3.2.8). Note that the analysis of the model skills (Sect. 3.1) is performed using this parameter set.

   2. More generic parameter sets depending on crop class and phenological stages only (named hereafter "pheno+cult")

   3. If no information is available for characterizing phenology, a calibrated set of parameters for the whole cultural crop
cycle is computed (hereafter named "culture only")

   4. The last case corresponds to the "optimal" parameter set but applied to another crop class in order to take into account potential errors that are likely to occur when a land-use map is used (named "unadapted").

What we consider the "best" case is very unlikely for a spatialized application of the tool because the largest the available database, it will never cover all the conditions encountered at the scale of an heterogeneous agricultural landscape where

each plot will have its specific soil, technical itinerary, hydric status, etc... Our objective is thus to get different parameter sets with value close to what is expected for each type of conditions (crops, climate, sites, phenological stages...) but without giving too much importance to the values themselves. To help perform the calibration, a stochastical multi-objective calibration method (Multi-Objective Calibration Iterative Procedure or MCIP; Demarty et al., 2004, 2005) has been implemented in order to minimize RMSE between simulations and measurements at half-hourly time intervals. Five target functions are identified

and minimized in the sense of Pareto for SEtHyS: $H$ and $LE$ fluxes, surface brightness temperature ($T_b$), net global radiation ($R_n$), and root zone soil water content (SWC). Considering its design based on remote sensing forcing, with less parameters, the evaluation of TSEB performances (Sect. 3.1) is sought in its out-the-box configuration presented in Norman et al. (1995). Nevertheless, a sensitivity analyse of the 3 parameters identified as the more sensitive for convection fluxes prediction by several studies including Diarra et al. (2017) is performed to evaluate the potential improvement. These 3 parameters are the

Priestley Taylor coefficient $\alpha_{PT}$, $\kappa$ the coefficient of net radiation extinction, and the empirical coefficient $\Gamma$ relating $R_{n[soil]}$ to $G$. Evaluation was performed for each parameter independently and with only two target functions ($H$ and $LE$).





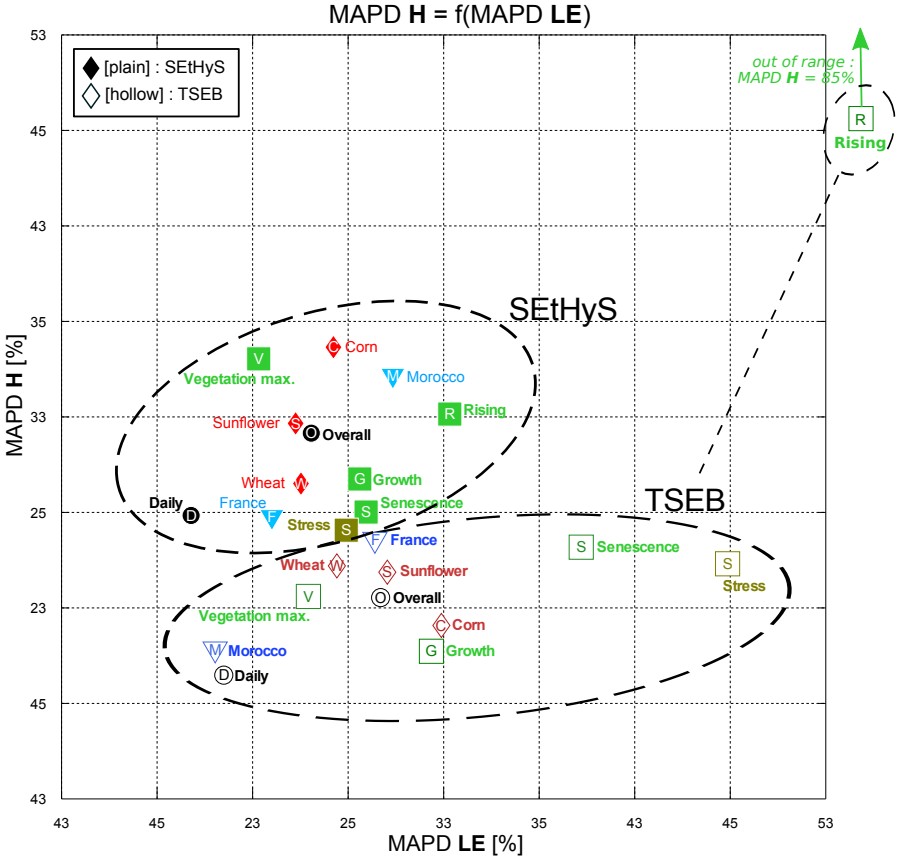

**Figure 2.** Comparison of TSEB (hollow markers) and SEtHyS (plain markers) estimations of $H$ and $LE$ for various time resolutions (circles), phenological stages (squares), cultures (diamonds) and climates (triangles). MAPD stands for Mean Absolute Percentage Deviation.

## 3 Results

### 3.1 Models skills by crops and phenological stages

Model simulations of heat fluxes are compared to tower fluxes measurements at half-hourly and daily time steps, with a focus on performance by crops and by growth stage. A specific paragraph in the discussion section is dedicated to the analysis of model's

5 behavior during stress periods. RMSE's for $LE$, $H$ and $R_n$ are displayed in Table 4 and MAPD's for $H$ and $LE$ are displayed in Fig. 2. Biases (not shown) are very limited and ranged between -23 $Wm^{-2}$ and +10 $Wm^{-2}$ for both models, except during the rising phase where they reach -47 $Wm^{-2}$ and +43 $Wm^{-2}$ for SEtHyS and TSEB, respectively (see discussion below). Available energy is well simulated for both models with daily averaged RMSEs of 43 $Wm^{-2}$ and 19 $Wm^{-2}$ for TSEB and SEtHyS, respectively. Regarding heat fluxes, Table 4 points out good performances of the TSEB model on daily averaged

10 values with regards to the relative simplicity of the approach compared to SEtHyS which relies on a systematic parameter





calibration. Both models exhibit close statistics concerning $LE$ estimations (RMSEs of 35.5 vs 38.9 $Wm^{-2}$ for SEtHyS and TSEB, respectively) while TSEB behaves slightly better regarding $H$ estimations (21.2 vs 28.7 $Wm^{-2}$). These values are close to errors found in the literature for TSEB (Norman et al., 1995; Zhan et al., 1996; Anderson et al., 1997; Kustas and Norman, 1999; French et al., 2005; Kalma et al., 2008; Diarra et al., 2013, 2017) and also within the range of expected errors from

EC towers measurements (Eugster et al., 1997; Wilson et al., 2002). Half-hourly values lead to similar conclusions, except that the drop in retrieved $LE$'s performance associated to this change of reference time interval is stronger for TSEB than for SEtHyS. Interestingly enough, this first analysis hides important disparities in terms of $LE$ prediction skills between the various growth stages. Indeed, Fig. 2 highlights some regularity in the SEtHyS skills regardless of growth stages and crops, as evidenced by the narrow group formed by the SEtHyS points. By contrast, the range of MAPD values for TSEB is much

wider. In particular, limitations of the model are clearly emphasized during rising and senescence stages. During the senescence phase, these discrepancies may both be attributed to stress (see discussion below) but could be related also to a poor partition of available energy between soil and vegetation. Indeed, the change in the radiative features of the canopy, including albedo, which occurs on senescent plants, is not taken into account by the model.

The poor performance during the rising stage is due to excessive limitation of the soil sensible heat flux, induced by the

parameterization of the roughness length for momentum ($Z0m = h_c/8$) at the denominator of the expression of the aerodynamic resistance $r_a$, leading to very high resistance when canopy height is very low. Since, during that stage, the vegetation net radiation is very limited, vegetation sensible heat is also close to 0. The observed high MAPD of $LE$ during the rising phase shall thus be attributed to significant bias of TSEB estimates. To a lesser extent, SEtHyS skills are also mitigated during the rising phase. Generally, when evaporation is predominant over transpiration, more weight is given to soil transfer processes

which are harder to characterize, considering the high heterogeneity of soil characteristics and the limited soil measurements available for calibration. The poor performances are more conspicuous with TSEB leading to estimation of $H$ with a MAPD of 85 %. By contrast, both models tend to have better performance when vegetation is fully-developed (MAPD less than 23 % for $LE$). The model performance by crop and growth stage is detailed in Fig. 3 (a) and (b), respectively, as normalized Taylor diagrams (Taylor, 2001). This diagram is a concise way to display the ratio between the variances of the model outputs

and the observed data, the correlation coefficient r and the RMSE between model estimates and observations normalized by the variance of the observed data set. The further from the point marked "observed reference" on the abscissa axis, the higher the normalized RMSE; likewise, dots on the right (left) side of the circle cutting the ordinate axis at "observed reference" overestimates (underestimates) the observation variance. Figures 3 (a) and (b) point out higher normalized standard deviations for TSEB $LE$ estimations. These noisier outputs are likely due to the instantaneous ("snapshot") computing architecture of

the model, while SEtHyS is better constrained by its continuous evolution of the soil water content which lead to smoother predictions of the daily cycle. This explains the drastic drops of TSEB RMSE on $LE$ when going from daily to half-hourly observations already underlined above. Finally, no significant skill differences are observed between crops, which seems to indicate that (1) the set of parameters used in TSEB describes well vegetation characteristics and that (2) the SEtHyS formalism can be adapted to various crops, provided that parameters are properly calibrated. More focus on the selected sets of param-

eters is given in the discussion section. Models performs well in both climate: SEtHyS showed slighty better performances





for flux estimates in France (MAPD for $LE$ of 23 % in France and 30 % in Morocco), whereas TSEB showed slightly better performances for flux estimates in Morocco (MAPD for $LE$ of 26 % in France and 18 % in Morocco). However, differences in crop management between France and Morocco and the availability of only one crop cycle in Morocco does not allows to draw final conclusions about climate impact on model skills. TSEB has low performances on senescence periods (including

hydric stress) for $LE$ estimation (MAPD of 45 %). This is partly due to Priestley-Taylor approximation which is suitable for unstressed vegetation in potential conditions (Priestley and Taylor, 1972). The other reason is also that it does not have water budget description. Increased LST resulting from water stress does not allow limiting $LE$ significantly enough in the TSEB scheme (see Sect. 3.2.7). Several authors have already pointed out that TSEB do not faithfully reproduce periods of senescence and water stress (Kustas et al., 2003; Crow et al., 2008; Boulet et al., 2015). SEtHyS includes description of soil water transfers

and leaf processes –in particular stomatal resistance– and can better reproduce hydric stress impact on $LE$ flux (MAPD of 28 %).

## 3.2 Sensitivity analyses to inputs and parameters

### 3.2.1 Overview

Given the overall purpose of our research dedicated to the spatialized estimation of evapotranspiration at various scales, quan-

tifying the decrease of model performance due to deterioration of input data quality combined with change of spatial scale from the field to a heterogeneous agricultural landscape is a prerequisite. The specific purpose of this section is twofold: (1) identify the most sensitive inputs and parameters and (2) quantify the expected model performances when realistic input errors are introduced. Uncertainties on input variables have been evaluated by comparing available *in situ* data to the spatialized datasets (SAFRAN meteorological reanalysis and ASTER, LANDSAT, FORMOSAT2 satellite imagery and products). Results

are presented on Table 1 and details are given in the following sections.

### 3.2.2 Intercomparison of SAFRAN and *in situ* meteorological data

Comparison results between the 2 available meteorological stations in the South-West France and the closest SAFRAN 8-kms grid points (inverse interpolated distance) are reported in Table 5 in terms of RMSE and biases (2006–2008 period). On average, SAFRAN behaves pretty well for air temperature and relative humidity with reasonable RMSE and biases close to 0. To a lesser

extent, wind speed is also well reproduced although slightly biased. The SAFRAN ability to predict incoming radiation is less convincing: bias is low but RMSE reaches 90 $Wm^{-2}$ (about 20 % on average). This comparison corroborates the conclusions of Quintana-Seguí et al. (2008) who also highlight a strong weakness of SAFRAN in terms of incoming radiation predictions. Er-Raki et al. (2010) used a forecast model (ALADIN from Météo-France) over the Tensift basin of Morocco. The results showed that the ALADIN forecasts are in good agreement with the station measurements in terms of solar radiation ($R_g$) and

air temperature ($T_a$). However, the comparison of the station and the forecasted values of relative humidity ($R_h$) and wind speed ($U_a$) are much more scattered. Besides the RMSE and biases representing time averaged statistical characteristics of the difference between SAFRAN and the two ground stations, it is also interesting to consider more extreme error values. To do so,





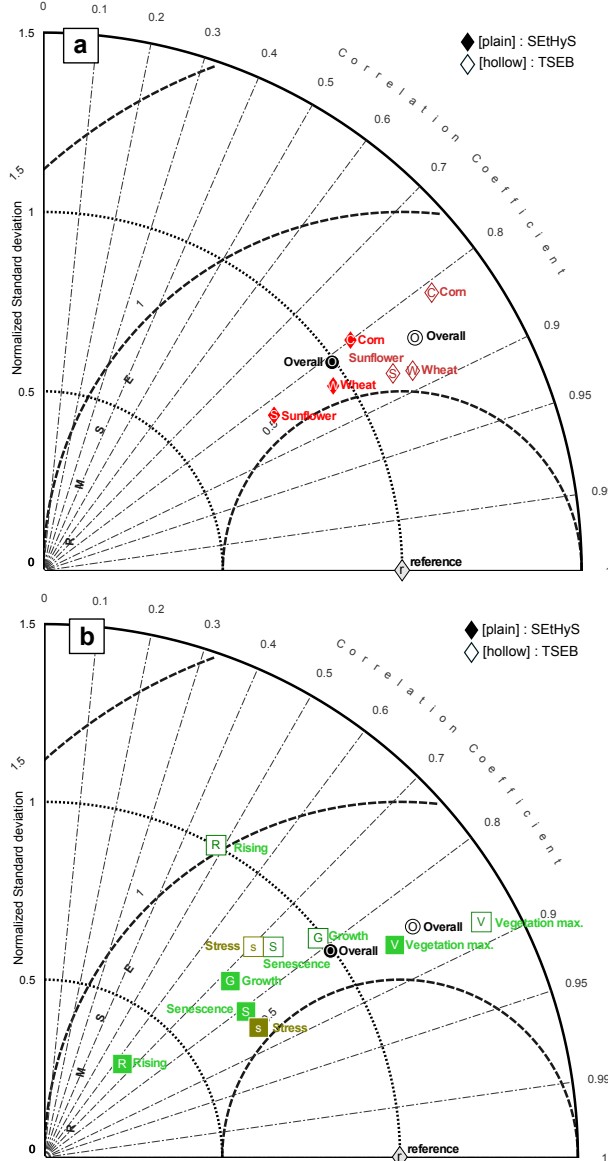

**Figure 3.** Taylor diagram for $LE$ performances of TSEB (hollow markers) and SEtHyS (plain markers), for various cultures (a) and phenological stages (b). Concentric lines centered on 0 indicate the normalized standard deviation with observations, radial lines indicate the correlation coefficient between simulations and observations, normalized RMSE isolines are the concentric circles centered on "r" (reference for the overall time series of observations: RSME=0, corr. coeff.=1 and NSD=1).

the $1^{st}$ and $9^{th}$ deciles of the difference distribution are shown in Table 5 in absolute values and in percentage. The probability of occurrence of such errors is far from insignificant as 20 % of the data are involved. These "extreme" errors are considered below for the sensitivity study regarding (1) the instantaneous estimates provided by the TSEB model depending on satellite





overpass time, leading to potential instantaneous errors much higher than the average; (2) the poorest quality of re-analysis data in the semi-arid areas because the meteorological station network may be scarcer.

### 3.2.3 Sensitivity analysis to meteorological inputs

Impact of these realistic and more extreme errors on convection fluxes simulations are shown in Fig. 4 and Fig. 5, respectively.
We will hereafter focus on noise since biases are often limited on re-analysis systems thanks to a system of bias reduction. On average, SEtHyS simulations are less sensitive to noisy inputs for $LE$ than for $H$, whereas reverse conclusions can be drawn for TSEB. Except for wind speed, adding noise to meteorological inputs has almost no impact on SEtHyS latent heat predictions, while noise added to incoming radiation and, to a lesser extent, air temperature, deteriorates TSEB predictions with RMSE of $LE$ simulations lowering from reference value of 55 $W m^{-2}$ to nearly 60 $W m^{-2}$. Indeed, whereas the partition
between latent and sensible fluxes is moderated by the slow-varying soil moisture content in SEtHyS, TSEB partition relies on measured available energy and surface temperature inputs only. By contrast, noisier wind speed, air temperature and, to a lesser extent, solar radiation, deteriorate significantly sensible heat for SEtHyS. TSEB appears, on average, less sensitive to noisy meteorological inputs for $H$. When considering extreme errors (Fig. 5) on meteorological forcing, the same variables are identified as the most sensitive ones: $R_a$, $R_g$ and $T_a$ for TSEB and $R_a$, $R_g$ and $R_h$ for SEtHyS. However, whilst SEtHyS
performance remains acceptable despite these high errors on forcing, TSEB performance for both $LE$ and $H$ collapse in response to incoming radiation errors in particular. Interestingly enough, incoming solar radiation can also be retrieved from satellite measurements such as MSG. In particular, a recent study by Carrer et al. (2012) points out a significant improvement of MSG derived short wave and long wave downwelling surface radiation with regards to the SAFRAN analysis system. This could represent a valuable alternative for regional assessment of evapotranspiration, particularly for the TSEB model. As a
conclusion, the SEtHyS model appears more stable to uncertain meteorological inputs than TSEB on average, at least for latent heat flux predictions.

### 3.2.4 Sensitivity analysis to vegetation forcing inputs

Focus here is put on evaluating the bias effect on SEtHyS and TSEB flux predictions. Indeed, on one hand, errors on vegetation characteristics are much more difficult to evaluate as *in situ* measurements are time-consuming and therefore not always
available at a small time interval. On the other hand, biases on satellite estimates are more likely to occur than white errors because of a detection limit of visible sensors in the case of sparse vegetation and a possible saturation effect when Leaf Area Index is above 3 $m^2\ m^{-2}$. On average, Claverie et al. (2011, 2012) highlight a potential bias of 20 % for $LAI$ estimated from FORMOSAT data. Canopy height ($h_c$) is not available directly from remote sensing data but can be estimated from $LAI$. Canopy height ($h_c$) was deduced from $LAI = f(h_c)$ relations, applying linear regression to each culture and phenological
stage available in our *in situ* data. This methodology provides estimations of $h_c$ with a MAPD of 30 %, and "extreme" bias up to 100 % (Bigeard, 2014). The results shown in Fig. 5 demonstrate that TSEB and SEtHyS sensitivity to bias on $LAI$ remains limited. By contrast, TSEB and, to a lesser extent SEtHyS, exhibit a much higher sensitivity to bias on canopy height ($h_c$) due to erratic transfer resistances when $h_c$ is too close to the height of the micrometeorological measurements. As $LE$ is computed




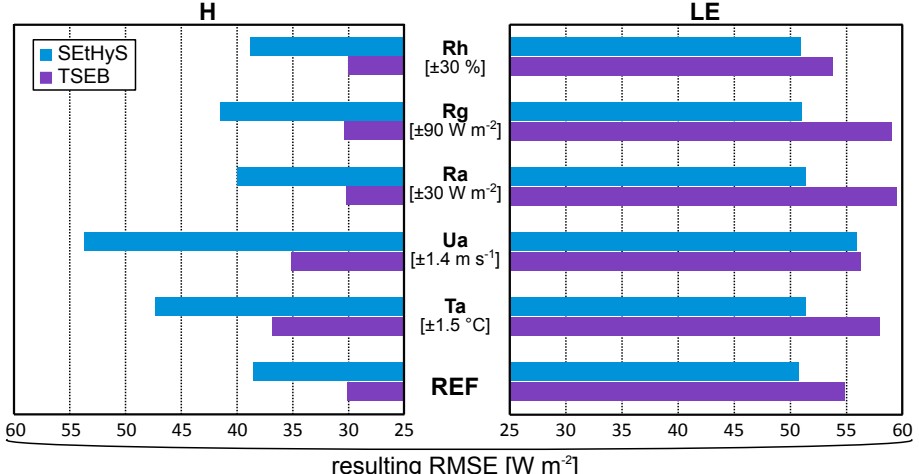

**Figure 4.** Sensitivity analysis (white noise) to meteorological inputs for both models and impact on estimation of $H$ & LE.

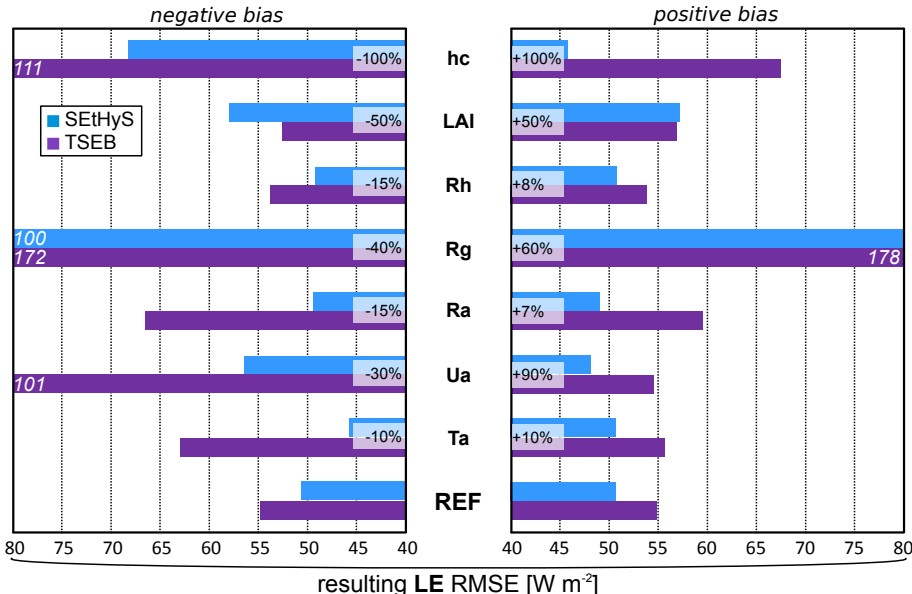

**Figure 5.** Sensitivity analysis to positive and negative biases applied to meteorological and vegetation inputs for both models.

from the residual of the energy budget in TSEB, a problem is observed on both $H$ and $LE$ fluxes, while $LE$ is less affected in SEtHyS (not shown).





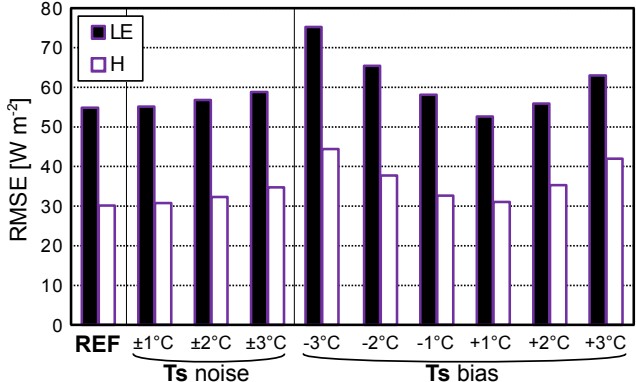

**Figure 6.** Sensitivity analysis to radiative temperature input for TSEB.

### 3.2.5 Sensitivity analysis to radiative temperature for TSEB

The comparison between *in situ* LST measurements and retrieval from the LANDSAT7 and ASTER images yielded a maximum absolute difference of 2.2 $K$ (4 points) in agreement with values reported in the literature ranging from 1 to 3°C (Hall et al., 1992; Gillespie et al., 1998; Schmetz et al., 2002; Peres and DaCamara, 2004; Li, 2004; Liu et al., 2006; Wan, 2008, among

others). As LST is expected to be a determining input of TSEB, an in-depth sensitivity analysis to this variable was carried out considering white noise and biases of 1, 2 and 3°C. Indeed, the spatial scale mismatch between the spatial sensor operating, at best, at 90m resolution and the SVAT model operating at the scale of an "agricultural unit" (potentially lower than a parcel) is likely to be important. Regarding the strong heterogeneity of agricultural landscape (in terms of crops, development stage, irrigation, hydric stress, etc.), bias is also likely to be important and quite impossible to correct. The results of adding errors to

measured radiative temperature on TSEB fluxes prediction are shown in Fig. 6. For limited white noise up to 2 $K$, the drop of TSEB skills is small on both $H$ and $LE$. By contrast, biases are much more impacting. In particular, a negative bias of 3 $K$ could deteriorate $LE$ RMSE from 58 $Wm^{-2}$ to 78 $Wm^{-2}$. Interestingly enough, a negative bias, that is likely to occur when the observed pixel is partly irrigated (i.e. cold), while the agricultural unit studied lay under stress (i.e. hot) for instance, has a stronger effect than a positive bias. This is likely to occur in many cases in practice: a mixed pixel including forest and stressed

field, irrigation heterogeneity within a pixel (for instance in progress irrigation within a field including gravity or center pivot system or the use of a localized sprinkler).

### 3.2.6 Sensitivity analysis to water inputs and soil water content for SEtHyS

Water inputs include rainfall and irrigation over agricultural landscape and both are difficult to assess accurately as long as the considered spatial scale exceeds one $km^2$. Even in this case, a good knowledge of irrigation input at the field level requires

costly field surveys, since farmers' associations or regional office responsible for irrigation water often work at a larger scale made of several plots. In addition to this potential uncertainty, the initial condition of soil water content (SWC) should also be considered uncertain as a result, for instance, from errors piling up from previous inputs. Figure 7 shows results of sensitivity



analysis to these three factors: uncertainty on irrigation amount and timing and on SWC initial condition. Unsurprisingly, all factors had a significant impact on $LE$ predictions. Even if input timing was correct, a bias of 1mm with correct initial SWC deteriorated the SEtHyS skill by 5 %. If the bias on input reaches 10mm and the initial SWC is negatively biased with the same level, the loss of model performance is above 25 %. Considering that the total amount of an irrigation round can reach

100mm, a 10mm uncertainty is very likely to occur in practice. In addition, a negative bias on SWC impacts significantly more $LE$ predictions than a positive bias. Indeed, going towards drier conditions may lead to stress and, as a consequence, to a drastic drop of predicted $LE$ compared to reference, whereas increasing SWC when the surface is already close to potential conditions won't have any effect on $LE$. Within this context, data assimilation of surface soil moisture retrieved from spatial sensors could provide an interesting solution to improve accuracy of SWC initial conditions (Prevot et al., 1984; Demarty

et al., 2005; Li et al., 2006). By contrast, the timing, although important, has a secondary influence on model skills. Even when water input is applied 3 days before or after the actual date, the loss of $LE$ predictions skills remain limited at around 15 %. Indeed, considering that agricultural landscape is often well-watered in order to maximize production, vegetation is able, through transpiration processes, to maintain high levels of $LE$ during long periods. The resulting dynamics of $LE$ is relatively smooth compared to bare soil that is dominated by evaporation processes. Finally, the main conclusion is that emphasis should

be laid on a water amount prescription whilst timing appears of secondary importance.

### 3.2.7 Cross sensitivity analyses of models through linkage of radiative temperature and SWC

Sensitivity of the TSEB and SEtHyS models to surface water status has to be detailed in order to compare how the models respond to a change in water conditions. The difficulty lies in the conceptual difference between both models: surface water status is an explicit variable state for SEtHyS while, in the TSEB model, surface radiative temperature is an indirect proxy of the

surface hydric conditions. For the set of simulation periods considered in this study, initial soil water contents (for surface and root zone) were biased in SEtHyS inputs with +/-10, +/-30 or +/-50 % levels. As a consequence, the simulated radiative surface temperature by SEtHyS diverges from reference and the differences between both temperatures simulations time-series are added to the TSEB model input radiative temperature as an equivalent water bias converted in temperature. It is assumed that the SEtHyS model, used with a calibrated set of parameters, is able to simulate a realistic temperature equivalent to the water

status biases (Coudert et al., 2006; Coudert and Ottlé, 2007). Figure 8 shows the average variation of the temperature bias as a function of the SWC bias. As expected, temperature increases with water content deficit. Beyond the [-10 % – +10 %] interval, temperature and water contents biases evolve quasi linearly with a greater increment for dry conditions. On the contrary, one can expect a more rapid limitation in temperature decrease with wet conditions, when soil reaches field capacity or saturation. The consequence on evapotranspiration deviation from reference clearly shows that beyond the [-10 % – +10 %] interval for

water content biases, the error increases also linearly with a greater increment for dry conditions. Under -20 % bias, the impact on $LE$ flux exceeds 50 $Wm^{-2}$. This result is important for our purpose to spatialize models for evapotranspiration estimates, because accurate root zone and surface water content retrievals from thermal and microwave remote sensing are a real challenge over heterogeneous landscapes (Barrett and Renzullo, 2009; Hain et al., 2011). The shift in temperature simulated by SEtHyS for -50 to +50 % water contents biases does not exceed 2 $K$ and lay therefore within the typical remotely sensed surface





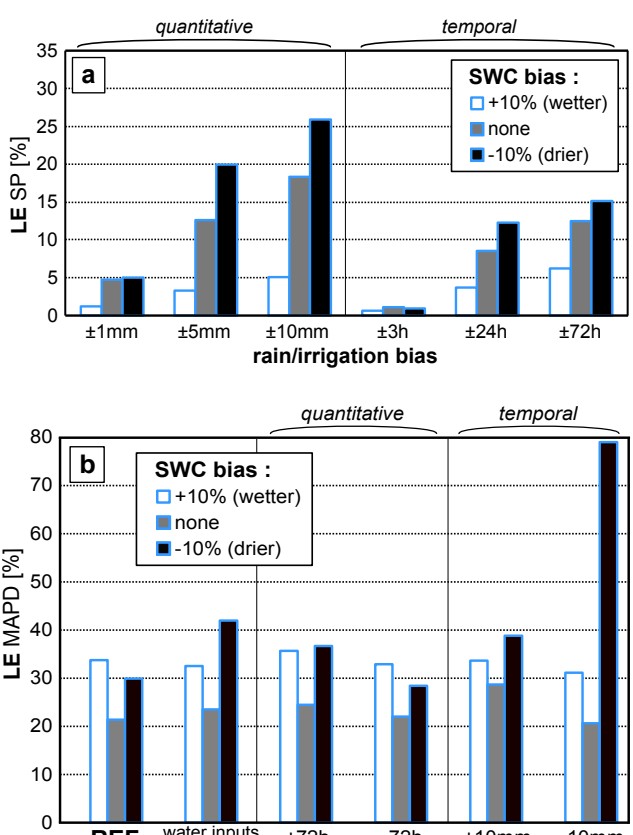

**Figure 7.** SEtHyS sensitivity analysis to rain and irrigation inputs, with influence of bias on Soil Water Content (SWC). SP in figure (a) is defined as $SP = \frac{|LE_{[positivebias]} - LE_{[negativebias]}|}{LE_{[reference]}}$.

temperature uncertainty range. For such a temperature bias, the TSEB model evapotranspiration divergence is lower than 40 $Wm^{-2}$. As a consequence, compared to the SEtHyS model, TSEB is less "reactive" to soil water contents variation. The result is critical for dry or stress conditions as previously pointed out. Actually, water status is only taken into account in the TSEB model through the surface temperature which is not sufficient and no additional limitation of surface evapotranspiration is done by modulating for instance the Priestley Taylor parameter.

### 3.2.8 Main results of parameters sensitivity analysis and calibration before landscape spatial distribution

The results of the parameters sensitivity analysis and calibration for each time period are described in parallel for both models and may be summarized as follows:

- The $V_{max0}$ values of the SEtHyS model are different for wheat (with $110 < V_{max0} < 140$ $\mu mol$ $m^{-2}$ $s^{-1}$) and for sunflower and corn ($65 < V_{max0} < 100$ $\mu mol$ $m^{-2}$ $s^{-1}$) at every time period on the french worksite. The SEtHyS





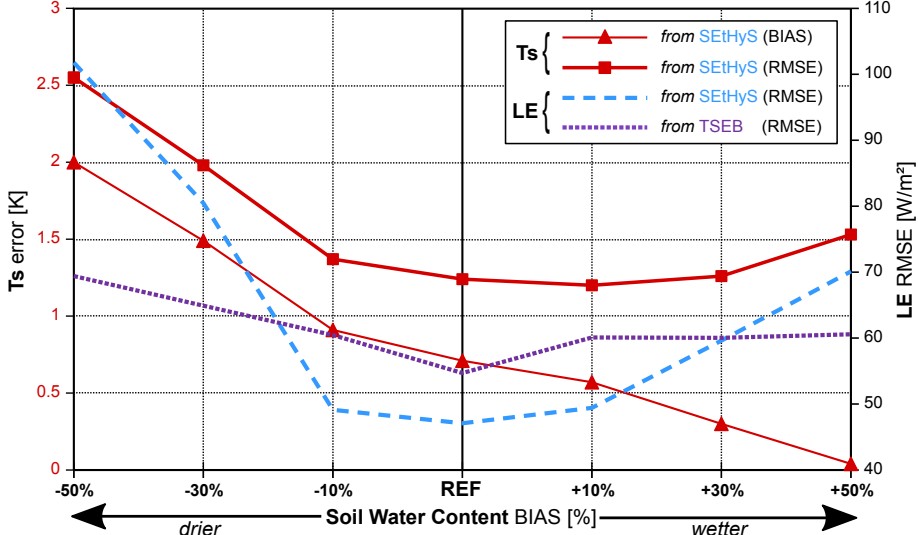

**Figure 8.** Error of Ts simulated by SEtHyS for increased and decreased SWC, and impact on models ETR estimates. TSEB is forced with Ts estimates from SEtHyS.

photosynthesis module was taken from the SiB2 model (Sellers et al., 1996b). Sellers et al. (1996b) suggested a value of $100 \, \mu mol \, m^{-2} \, s^{-1}$ for the "agriculture biome" land-cover class that is in-between our calibrated values. This means that the photosynthesis assimilation rate has to be enhanced for the wheat class in order to match the set of target variables ($H$, $LE$, $w_2$, $R_n$, $aR_g$). This result is consistent with the higher optimal value also obtained for the TSEB $\alpha_{PT}$ parameter (Priestley and Taylor, 1972; Norman et al., 1995) for wheat ($\simeq 1.5$) than for corn and sunflower ($\simeq 1.35$). A more detailed

discussion about this parameter is carried out in the following section. Measurement of photosynthesis assimilation rate and stomatal conductance were carried out over wheat, corn and sunflower crops with a Licor 6400 gas analyzer in 2010 within the framework of the "SudOuest" project. Optimal values deduced from measurements for the $V_{max0}$ parameter (not shown) are in perfect agreement with the previous calibration results, except for sunflower. We should suspect

overestimated model inputs of $LAI$ based on destructive measurements due to the sunflower heterogeneity at field scale, leading to $V_{max0}$ parameter values that are lower with calibration than expected with measurements.

  – The calibrated soil characteristics of the SEtHyS model ($W_{resid}$ and $W_{max}$) closely follow the type of soil identified at the various experimental sites. $W_{max}$ parameter is lower for Auradé than for Lamasquère because of lower clay content, but a noticeable evolution are found with plant growth. Wmax is high or maximal at plant emergence or early growth

especially for wheat and then decrease with plants growth and associated soil compaction. Similarly, the $W_{resid}$ soil parameter, in particular for sunflower, gradually increases from plant emergence ($0.07 \, m^3 \, m^{-3}$) to maximal growth ($0.14 \, m^3 \, m^{-3}$) and then decreases with senescence ($0.08 \, m^3 \, m^{-3}$). As a consequence, $W_{max}$-$W_{resid}$ decreases with plant growth. Calibration results for root zone depth $dp_2$ show that values obtained during emergence or early growth are higher than expected, except for wheat crop in Morocco. This means that, from a model calibration point of view, available water





storage capacity (depending on $W_{max}$-$W_{resid}$ and $dp_2$) is not limiting for French sites during the beginning of the crop cycle, but it becomes a constraint when water deficit grows between spring and summer. The hydraulic conductivity at saturation $K_{sat}$, exhibits also the lowest values (< 2.10-7 $.s^{-1}$) for the Lamasquère site with higher clay content and lower sand content (Genuchten, 1980), compared to Auradé and Morocco sites. For all sites the $K_{sat}$ minimal values are reached during plant emergence stages and generally increase during growing periods. This result is also consistent with the root system development favoring the vertical soil water transfers.

– The $F_{therm}$ parameter is a correction coefficient of the volumetric soil heat capacity which allows limiting the ground heat conduction flux in the "force-restore" soil system. This parameter is one of the most influential parameters of the SEtHyS model. For all sites, $F_{therm}$ is high with 1.5 to 1.7 values for plant emergence and early growth and then decreases as vegetation grows to its maximal stage of development, to reach its lowest value (nearly 1). This result is also consistent with the development of the root system which grows in the upper soil horizon at plant emergence, increasing the soil porosity in agreement with the result obtained for the $W_{max}$ parameter. Then, the soil compacts, which leads to lower porosity and higher heat capacity. A similar result is obtained with the TSEB model. Choudhury et al. (1987) proposed a value of 0.35 for the $\Gamma$ parameter (see Eq. 10) while higher values ($\simeq$0.5) gives better results after calibration at maximal vegetation development periods, increasing heat conduction flux in the soil (see Fig. 9 (a)). The impact on the $H$ and $LE$ convective flux is nevertheless fairly low (less than 5 $Wm^{-2}$ for averaged RMSE).

– Finally, the default value of 0.45 for $\kappa$, the coefficient of the exponential extinction of net radiation to compute $R_{n[soil]}$ (Eq. 9), appears generally acceptable for $H$ and $LE$ predictions except for sunflower (see Fig. 9 (b)). Indeed, as far as sunflower is concerned, the hypothesis of spherical leaf angles distribution is less relevant even for the simple radiative transfer model included in the models, based on a turbid description of the canopy. A value close to 0.65 is more optimal according to calibration, especially for periods around maximal $LAI$. Such higher value reduce the net radiation in the soil ($R_{n[soil]}$ in Eq. 9).

– Concerning climate differences between French and Moroccan sites, both models exhibit a coherent trend to limit evapotranspiration over the semi-arid site. For TSEB, this is achieved by lowering the $\alpha_{PT}$ parameter (wheat optimal value of 1.25 for Morocco against 1.6 for France) which directly influence the evapotranspiration rate, while SEtHyS represents this limitation by lowering the photosynthesis activity, which is controled thru $V_{max0}$ parameter (optimal values between 40 and 70 $\mu mol\ m^{-2}\ s^{-1}$).

– The temporal evolution of the surface albedo retrieved by calibration is also interesting. As vegetation grows, the calibration results show a slight increase of the vegetation layer albedo and a decrease just after senescence phase before harvest. Around maximal $LAI$, winter wheat albedo for the French sites obtained from calibration ($\simeq$0.3) is slightly higher than sunflower or corn albedos ($\simeq$0.25). Despite these overestimated values, the experimental and bibliographical values (0.22–0.23), tends to corroborate the experimental observations of daily mean albedo based on Plant Area Index presented by Ferlicoq et al. (2013).





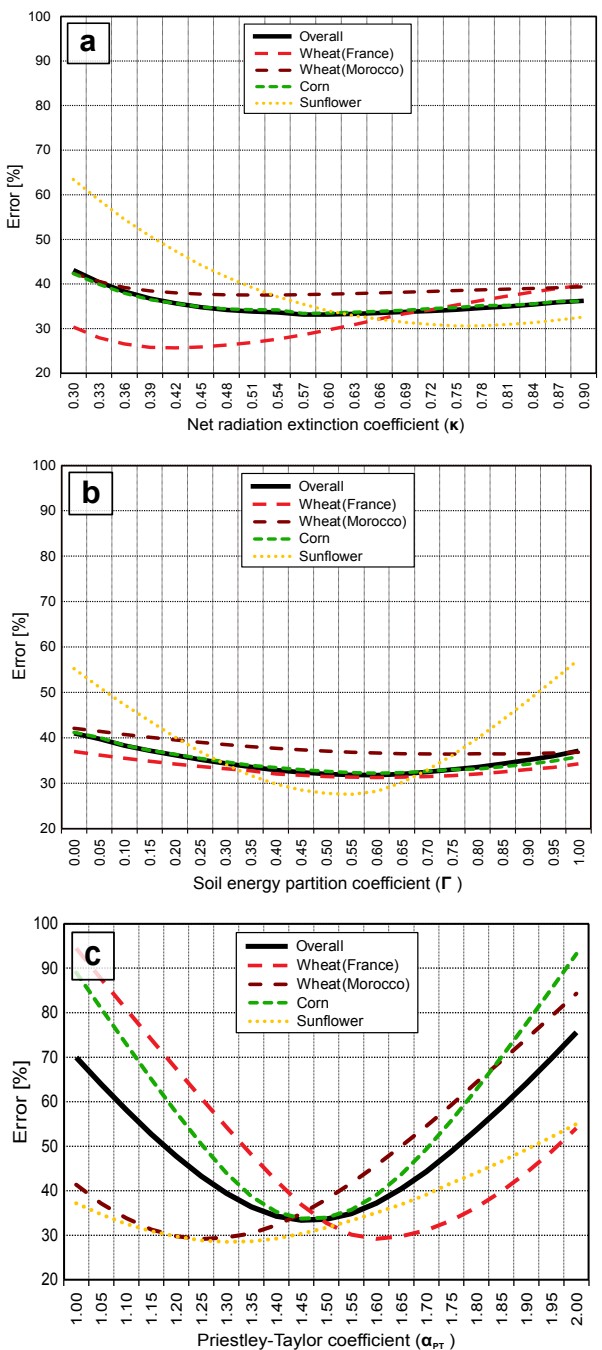

**Figure 9.** Sensitivity analyses of $\Gamma$ (a), $\kappa$ (b) and $\alpha_{PT}$ (c) TSEB parameters during vegetation periods. Error was computed as a cost function (euclidian distance) taking into account MAPD of $LE$ and $H$ simultaneously: $Error = \sqrt{MAPD_{[LE]}^2 + MAPD_{[H]}^2}$.





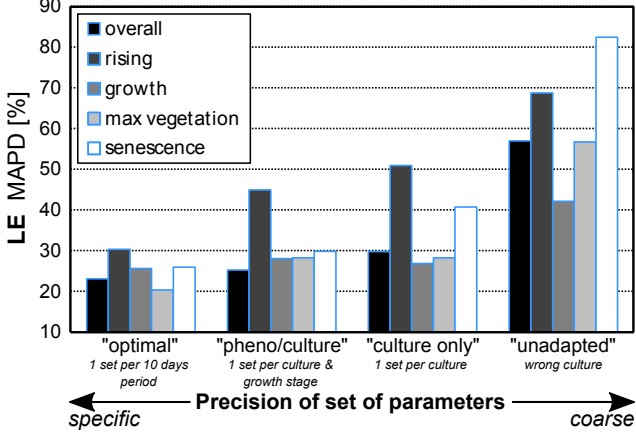

**Figure 10.** Impact of precision of set of parameters on simulated fluxes ("optimal": most accurate per 10days period), "pheno+culture.": per culture and phase of growth as used for spatialization, "culture only": per culture (when only soil occupancy is known), "unadapted": incoherent (wrong culture)).

## 4 Discussion

### 4.1 Influence of the parameters sets for model spatialization

The four calibration cases for the SEtHyS model going from site and period specific to more generic parameters from the literature are considered in order to evaluate the potential loss of model performance when specific calibration is not possible

by lack of data. Figure 6 shows the impact of the parameter set used on the SEtHyS performance to predict $LE$ fluxes. Global results (for all crop classes and the whole cultural cycles) corresponding to the label "*overview*" in Fig. 6 give a MAPD of 30 % for the generic "*culture only*" set of parameters. This result does not differ much from the performance obtained with more specific sets of parameters "*pheno+cult*" or "*optimal*" giving 25 % and 23 % of MAPD, respectively. However, when a set of parameters from another crop class is used, MAPD reaches 58 %. A finest analysis by phenological stages indicates an overall

stability of the results with the "*pheno+cult*" parameter set with regards to "*culture only*". There are actually two exceptions: one for the vegetation senescence periods which require specifics parameter sets. A mean set of parameters for the crop class increases MAPD from 30 % to 40 %. The second relates to crop rising periods. A generic one based only on the crop class ("*culture only*") increases MAPD up to 50 % compared to 45 % for "*pheno+cult*" when taking into account the phenology. As a conclusion, a mean parameter set associated to a specific crop without considering phenology implies only a slight decrease

of the performance for growth or maximum vegetation development. By contrast, the relevance of the parameter sets becomes noticeable when specific information is not available for rising and senescence periods (including potentially water stress phases). With the same purpose, a specific analysis is dedicated to the Priestley-Taylor $\alpha_{PT}$ key parameter of the TSEB model in the next section.



## 4.2 A deeper look at the $\alpha_{PT}$ parameter for spatialization

A first estimation of $LE_{vegetation}$ canopy transpiration flux is obtained from the Priestley-Taylor approximation and depends on the fraction of green $f_g$ and on the $\alpha_{PT}$ parameter. Most studies (Norman et al., 1995; Kustas and Norman, 1999; French et al., 2003; Anderson et al., 1997, 2008; Li et al., 2006, 2008, among others) have usually used a $\alpha_{PT}$ value of about 1.3

for semiarid or sub-humid agricultural areas. However, this value may vary with vegetation type as mentioned in Norman et al. (1995), low values of $LAI$, atmospheric demand (Anderson et al., 2008; Agam et al., 2010; P. D. Colaizzi, N. Agam, J. A. Tolk, S. R. Evett, T. A. Howell and S. A. O'Shaughnessy, W. P. Kustas, 2014) or dry air advection conditions (Kustas and Norman, 1999). As a first step, the calibration is performed for midday time interval series over various surface and atmospheric conditions in order to be compared with previous studies using TSEB instantaneously for water flux mapping purpose when

thermal imagery is available. Figure 9 (c) shows the influence of $\alpha_{PT}$ values on $H$ and $LE$ fluxes for wheat, corn and sunflower crops over the sites in both the South West of France and Morocco. Optimal values for irrigated wheat in Morocco (semi-arid climate) and sunflower in the South-West of France (temperate climate) are close to the 1.3 bibliographical value. For wheat and irrigated corn in South-West of France, mean optimal values are higher and reach 1.6 for wheat. Mean optimal value of 1.5 is obtained for temperate climate, while a lower value of 1.25 is obtained for semi-arid climate. In a second step, the half-hourly

data are used for the calibration in order to study the diurnal cycle of the $\alpha_{PT}$ parameter. The $\alpha_{PT}$ parameter shows a U-shape diurnal cycle evolution as displayed in Fig. 11 with smaller values around midday time, and higher values in both morning and evening when stability conditions are changing, enhancing $LE_{vegetation}$ transpiration canopy flux. This is particularly outlined under clear sky conditions, when TIR data from space is most likely to be collected. The original $\alpha_{PT}$ parameter is defined for a system at equilibrium with constant temperature, a condition which is particularly not met in the morning and

in the evening when temperature temporal gradients are the highest. As a consequence, such variations integrated over the diurnal cycle lead to slightly higher $\alpha_{PT}$ fixed optimal values for daily half-hourly time interval simulations. Moreover, results indicate a decrease of RMSE by about 10 % on both $H$ and $LE$ fluxes when optimal values at the original time interval are used instead of a fixed daily average. Nevertheless, as more mistakes on fluxes estimation are likely to be made around midday time, when turbulent fluxes are maximal, optimal daily value of $\alpha_{PT}$ tends towards its value around mid-day and is not much

affected by increased morning and evening values. Despite thermal imagery from space is not available with the presence of clouds, the emergence of drone acquisition makes interesting the characterization of $\alpha_{PT}$ under those conditions. On cloudy days, Fig. 11 hilights that fixed daily optimal values of 1.8 to 2 (higher instantaneously) are required for optimizing $H$ and $LE$ fluxes enhancing again the $LE_{vegetation}$ transpiration flux for such reduced atmospheric demand. Hence, for simulation under cloudy conditions, values can be raised by +0.4 in a view to interpolate time series between satellite overpass or to run TSEB

model with *in situ* or low altitude aircraft remotely sensed surface temperature. An improvement of about 10 % on $LE$ flux simulation is likely to be expected when taking into account the above-mentioned impact of vegetation and cloudy conditions considerations on $\alpha_{PT}$ parameter retrieval. However, recently, P. D. Colaizzi, N. Agam, J. A. Tolk, S. R. Evett, T. A. Howell and S. A. O'Shaughnessy, W. P. Kustas (2014) remembered that larger $\alpha_{PT}$ values did not mitigate the discrepancies on the evaporation (E) and transpiration (T) components of the total latent heat flux (ET). These authors have proposed a revised





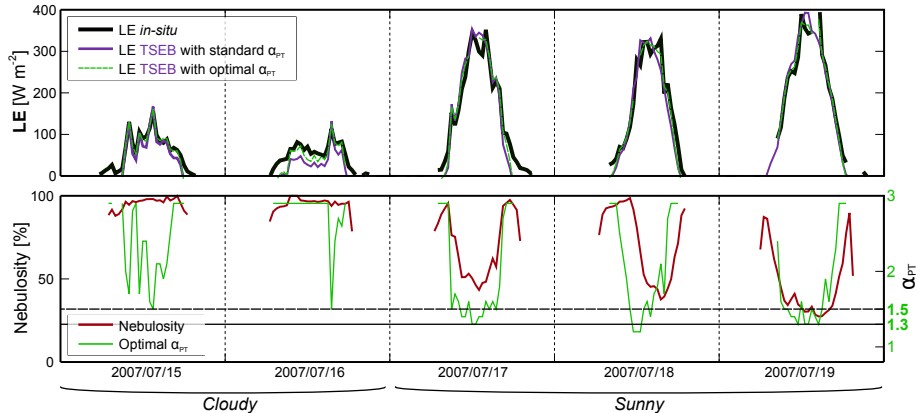

**Figure 11.** Influence of using optimal $\alpha_{PT}$ or averaged $\alpha_{PT}$ on $LE$ estimates by TSEB, and influence of nebulosity on optimal $\alpha_{PT}$ estimates. Nebulosity was computed using Photosynthetically Active Radiation (PAR) measurements: $Nebulosity = \frac{PAR_{[diffuse]}}{PAR_{[total]}}$. Sample from Auradé 2007 sunflower plot.

version of TSEB replacing the Priestley-Taylor formulation with the Penman-Monteith equation in order to better account for large variations of vapor water pressure deficits and correct the evaporation, transpiration and total $LE$ simulations. Boulet et al. (2015), thus built the SPARSE model based on Penman-Monteith with satisfying performances with the Morocco wheat site dataset, above those of TSEB with default parameter values.

## 5 Conclusions

This study aimed at comparing a "complex" SVAT model with the instantaneous energy balance model TSEB with the objective to map distributed evapotranspiration on agricultural landscapes at different resolutions. This study focused specifically on the comparison of the TSEB model (Norman et al., 1995) with the SEtHyS SVAT model (as described in Coudert et al., 2006) with two main objectives: (1) the accuracy assessment through the comparison of model predictions to a large *in situ* database; (2) a comprehensive sensitivity analysis to uncertainties in inputs and parameters of both models potentially induced by landscape spatialization. The main results of the study can be summarized as follows: Both models give close statistics on daily average $LE$ (RMSEs of 36 vs 39 $Wm^{-2}$ for SEtHyS and TSEB, respectively) while TSEB behaves slightly better regarding $H$ estimations (21 vs 29 $Wm^{-2}$). This points out remarkable performance of the TSEB model compared to the relative simplicity of the approach, all the more given that SEtHyS parameters are calibrated for each cases considered (crop, phenology and site). Nevertheless, SEtHyS skills appear more stable regardless of growth stages and crops whilst limitations of the TSEB model are clearly emphasized during rising and senescence stages. SEtHyS simulations are also less sensitive to noisy meteorological inputs for $LE$ than TSEB, for which performance are significantly deteriorated by noise on meteorological data, in particular for the incoming radiation. Indeed, the partition between latent and sensible fluxes is moderated by the slow-varying soil moisture content in SEtHyS, while the TSEB partition relies on measured available energy and surface temperature inputs





only. When considering extreme but likely errors on meteorological forcing, TSEB skills for both $LE$ and $H$ explode (still in response to incoming radiation errors in particular). TSEB and SEtHyS sensitivity to bias on $LAI$ remains limited, but TSEB exhibits a much higher sensitivity to bias on canopy height ($h_c$) which is due to erratic transfer resistances when h is too close to the height of the micrometeorological measurements. The sensitivity analysis on surface temperature which is one of the

more important inputs for TSEB shows that for a limited white noise up to 2 $K$, drop of TSEB skills is small on both $H$ and $LE$. By contrast, biases are much more impacting as a negative bias of 3 $K$ could deteriorate $LE$ RMSE from 58 $Wm^{-2}$ to 78 $Wm^{-2}$. Similarly, the sensitivity of SEtHyS skills to uncertain water inputs and initial soil water content, specific inputs of the SEtHyS model, has also been analyzed separately. We show that emphasis should be laid on water amount retrieval whilst timing of water supply appears of secondary importance; in particular a 10mm negative bias on input coupled to a negatively

biased initial SWC of 10 % with the same level, lead to a the loss of model performance above 25 %. A cross sensitivity analysis of the TSEB and SEtHyS models to surface water status was carried out as follows: several surface temperature time series are simulated by SEtHyS with initial soil water contents bias by +/-10, +/-30 or +/-50 %. The difference of surface temperature compared to a reference simulation is added as input to the TSEB model as an equivalent water bias converted in temperature. The shift in temperature simulated by SEtHyS for -50 to 50 % water contents biases does not exceed 2 $K$

and is therefore within the typical remotely sensed surface temperature uncertainty range. For such a temperature bias, the TSEB model evapotranspiration divergence is lower than 20 $Wm^{-2}$ while it reaches 50 $Wm^{-2}$ for SEtHyS. As a conclusion, TSEB is less "reactive" to soil water contents variation than the SEtHyS model. The sensitivity analysis and calibration study of the models for various crop, phenology, surface conditions and atmospheric forcing combinations can be used as a learning basis for model spatial distribution at the landscape scale of a small region. After checking that calibrated parameters are

consistent with biophysical processes governing the hydrological functioning of the studied crops, several parameters sets for both models are considered, from site-specific, crop and phenology to average set for one crop, and skill losses have been evaluated. We show that an average parameter sets for a specific crop, without considering phenology, implies only a slight decrease of the performance of SEtHyS for growth and maximum vegetation development stages. By contrast, the relevance of the parameters sets becomes noticeable when specific information is not available for rising and senescence periods (including

potentially water stress periods). For TSEB, the most sensitive parameter is the Priestley-Taylor coefficient $\alpha_{PT}$. Higher values than the literature have been highlighted in this study, in particular for cloudy conditions: optimal values range from 1.3 to 1.6. In addition, a more in-depth analysis points out a "U-shaped" diurnal cycle evolution of this parameter, with smaller values around midday time and higher values in the morning and the evening. This is particularly outlined under clear sky conditions. As a conclusion, this result completes the study of Crow et al. (2005) comparing TSEB spatialized fluxes with the

TOPLATS SVAT simulations (Famiglietti and Wood, 1994; Peters-Lidard et al., 1997), giving an edge to the use of TSEB when models are spatially distributed with necessary sparse or uncertain observations (rainfall, soil characteristics, etc...). The overall performance of the TSEB model are particularly remarkable considering the lower number of parameters and, in particular, that there is no need for water input knowledge compared to a more complex SVAT model. Nevertheless, the more in-depth sensitivity analysis points out specific conditions when performance can be very poor: during emergence and senescence phase,

and when incoming solar radiation is of bad quality. SEtHyS appears much more stable with regards to uncertain inputs but



require a larger number of parameters. This study lays the foundation for the spatialized application of both models at different

resolutions, when thermal infrared data are available for ETR estimation purposes. Based on this study, our further work deals

with 1) developing a methodology taking advantage of remote sensing spatial surface temperature contrasts at landscape scale

for different resolutions and 2) better documenting crop phenological cycles with multi-sources radar data (Fieuzal et al., 2012,

2013) in the SEtHyS model landscape spatialization. The first perspective is carried out within the frame of the preparation of

the future TRISHNA mission of the French space agency (CNES) and the Indian Space Research Organization (ISRO), and

the second one will take advantage of the recent SENTINEL Mission of the European Space Agency (ESA).

*Data availability.*   Data access from the French and the Moroccan sites must be requested to the head of the Sud-Ouest observatory (Tiphaine

Tallec, CESBIO, France) and to the head of the TENSIFT observatory (Jamal Ezzahar, UCAM, Morocco). The SAFRAN data should be

directly requested to the head of Météo-France (Toulouse, France).

## Appendix A:  SEtHyS main equations

This section presents the governing equations for the SEtHyS SVAT model variables.

### A1    Basical set of equations for the SEtHyS model

The mass and energy budget is solved jointly for both soil and vegetation sources from the following system:

$$
\begin{cases}
R_{n[soil]} = H_{[soil]} + LE_{[soil]} + G \\
R_{n[veg]} = H_{[veg]} + LE_{[veg]} \\
H = H_{[veg]} + H_{[soil]} \\
E = E_{[veg]} + E_{[soil]},
\end{cases}
\tag{A1}
$$

where $R_{n[soil]}$ and $R_{n[veg]}$ are net radiations at soil and vegetation levels and $G$ is the soil heat flux. Parameterization of the

soil behavior is based on Deardorff's formalism (1978). The soil surface temperature $T_{[soil]}$, the vegetation temperature $T_{[veg]}$,

the air temperature inside the canopy $T_{[canopy]}$ and the air humidity inside the canopy $q_{[canopy]}$ are determined by a first order

linearization of the previous equations system.

The soil surface temperature method prediction is namely the force restore method (Bhumralkar, 1975; Blackadar, 1976) and

requires deep soil temperature $T_2$. $T_2$ can be estimated from the mean air temperature over the 24 previous hours for short-

range studies (Blackadar, 1976). The heat capacity is prescribed by de Vries's model (1963) and hydrodynamic properties result

from pedotransfer functions (retention curve, hydraulic conductivity) based on Genuchten's approach (1980) under Mualem



hypothesis (1976).

Prognostic equation for ground surface temperature is written as:

$$\frac{\partial T_{[soil]}}{\partial t} = \frac{2\sqrt{\pi}}{C_e}(R_n - H - LE) - \frac{2\pi}{\tau}\left(T_{[soil]} - T_2\right). \tag{A2}$$

The factor $C_e$ is an equivalent heat capacity related to the diurnal thermal wave damping layer. In SEtHyS, the parameteri-
zation of the equivalent heat capacity has been weighted by introducing an empirical factor ($F_{therm}$ in parameters list, Table
2) compared to Deardorff (1978).

Deardorff (1978) proposed a similar approach of ground soil moisture, leading to the following equations:

$$\begin{aligned}
\frac{\partial w_g}{\partial t} &= -\frac{E_g + 0.2E_v\left(\frac{w_g}{w_{\max}}\right) - P}{dp_1} \\
&\quad - C\left(w_g, w_2\right)\left(w_g - w_2\right) \\
\frac{\partial w_2}{\partial t} &= -\frac{E_g + E_v - P}{dp_2},
\end{aligned}$$
(A3)
(A4)

where $w_{\max}$ is the soil moisture at soil saturation, $w_g$ and $w_2$ are surface and root zone water contents, $P$ is the precipitation
rate, $dp_1$ and $dp_2$ are the surface and root zone layers depths.

## A2 Radiative budget

Incoming radiation partition for optical (VIS) and infrared (IR) wavelength is performed through a shielding factor $\sigma_f$ tighly
linked to vegetation density. Its expression is as follows by considering a spherical distribution of leaves (François, 2002) with
the hypothesis of diffuse radiation for longwave domain and direct vertical radiation in shortwave domain:

$$\begin{cases}
\sigma_f = 1 - e^{-0.825LAI} & \text{for longwave domain} \\
\sigma_f = 1 - e^{-0.5LAI} & \text{for shortwave domain}
\end{cases} \tag{A5}$$

Radiative budget is then solved jointly at the soil and at the vegetation level for short and long wavelengths. Concerning short
wavelengths, soil albedo $\alpha_{soil}$ is linearly linked to surface soil moisture. Vegetation albedo $\alpha_{veg}$ is a model parameter. The net
radiation for the soil $R_{n[soil],SW}$ and for the vegetation $R_{n[veg],SW}$ are as follow ("Mod3" parameterization as proposed in
François, 2002):

$$R_{n[soil],SW} = S^{\downarrow}\frac{(1 - \sigma_f)(1 - \alpha_{soil})}{1 - \sigma_f\alpha_{veg}\alpha_{soil}}, \tag{A6}$$



and at canopy level:

$$R_{n[veg],SW} = S^{\downarrow}(1 - \alpha_{veg})\sigma_f \left[ 1 + \alpha_{soil} \frac{(1 - \sigma_f)}{1 - \sigma_f \alpha_{soil}\alpha_{veg}} \right] \tag{A7}$$

where $S^{\downarrow}$ is the incoming shortwave radiation.

Concerning long wavelengths, the net radiation for soil $R_{n[soil],LW}$ and vegetation $R_{n[veg],LW}$ are given by:

$$R_{n[soil],LW} = (1 - \sigma_f)\frac{\varepsilon_g(R_a^{\downarrow} - \sigma T_{[soil]}^4)}{1 - \sigma_f(1 - \varepsilon_f)(1 - \varepsilon_g)}$$
$$- \frac{\varepsilon_g \varepsilon_f \sigma_f \sigma(T_{[soil]}^4 - T_{[veg]}^4)}{1 - \sigma_f(1 - \varepsilon_f)(1 - \varepsilon_g)} \tag{A8}$$

$$R_{n[veg],LW} = \sigma_f \left[ \varepsilon_f(R_a^{\downarrow} - \sigma T_f^4) + \frac{\varepsilon_g \varepsilon_f \sigma(T_{[soil]}^4 - T_{[veg]}^4)}{1 - \sigma_f(1 - \varepsilon_f)(1 - \varepsilon_g)} \right]$$
$$+ \sigma_f \frac{(1 - \varepsilon_f)(1 - \varepsilon_g)\varepsilon_f(R_a^{\downarrow} - \sigma T_{[veg]}^4)}{1 - \sigma_f(1 - \varepsilon_f)(1 - \varepsilon_g)} \tag{A9}$$

Direct solar shortwave radiation $S^{\downarrow}$ and atmospheric longwave radiation $R^{\downarrow}$ are input model data.

The thermal infrared surface temperature $T_B$ (observed above the canopy) results from the partitioning of the surface and the radiative interaction between soil (whose temperature is $T_{[soil]}$) and the vegetation above (whose temperature is $T_{[veg]}$).

**A3    Heat fluxes expressions**

The mass and energy transfers in equilibrium with net surface radiation are momentum, sensible and latent heat fluxes. A conductance formalism allows expressing them by considering the canopy as a single vegetation layer (at some height $Z_{af}$)

above ground (Thom, 1972). Thus, following the electrical (Ohm's law) analogy, soil surface, leaves surface, air canopy space and atmosphere above canopy are the levels between which differences of potential (temperature and humidity gradients) and transfer coefficients *i.e.* aerodynamic conductances can be calculated.

Heat fluxes $H$ and $LE$ (sensible and latent heat fluxes respectively) are then determined at three levels:

at atmospheric reference level,

$$H = \rho c_p C_h \left( T_{[canopy]} - T_a \right) \tag{A10}$$

$$LE = \frac{\rho c_p}{\gamma} C_h \left( q_{[canopy]} - q_a \right) \tag{A11}$$



at vegetation level,

$$H_{[veg]} = \rho c_p C_{h[veg]} \left( T_{[veg]} - T_{[canopy]} \right) \tag{A12}$$

$$LE_{[veg]} = \frac{\rho c_p}{\gamma} C_{h[veg]} R' \left( q_{sat}(T_{[veg]}) - q_{[canopy]} \right) \tag{A13}$$

and at ground level,

$$H_{[soil]} = \rho c_p C_{h[soil]} \left( T_{[soil]} - T_{[canopy]} \right) \tag{A14}$$

$$LE_{[soil]} = \frac{\rho c_p}{\gamma} C_{h[soil]} C_s \left( q_{sat}(T_{[soil]}) - q_{[canopy]} \right) \tag{A15}$$

with

$$LE = LE_{[soil]} + LE_{[veg]} \tag{A16}$$

$$H = H_{[soil]} + H_{[veg]} \tag{A17}$$

and $G$ conduction heat flux in soil is residual of the energy budget :

$$G = R_{n[soil],LW} + R_{n[soil],SW} - H_{[soil]} \smallsmile LE_{[soil]} \tag{A18}$$

where $C_p$ is the specific heat at constant pressure, $\gamma$ is the psychrometric constant, $T$, $q$ are temperature and water vapor pressure and $a$, $g$, $canopy$ are indices relative to air, ground, and canopy air space.

$C_h$, $C_{h[veg]}$ and $C_{h[soil]}$ are respectively aerodynamic conductances between canopy air space and the overlaying atmosphere,

leaves surface and canopy air space, ground and canopy air space, $R'$ factor is defined below. These variables are derived from the eddy fluxes theory between two atmospheric levels. In SEtHyS model, the formulation follows the parameterization proposed by Shuttleworth and Wallace (1985) with a constant extinction coefficient in the exponential wind speed profile.

$C_s$ is the ground evaporation conductance; it depends on soil moisture conditions and potential evaporation $E_{pot[soil]}$ (Bernard et al., 1986; Wetzel and Chang, 1988; Soares et al., 1988):

$$C_s = \min \left( 1, \frac{E_{\lim}}{E_{pot[soil]}} \right), \tag{A19}$$





where $E_{lim}$ depends on soil properties (composition and moisture), Soares et al. (1988) gives the expression:

$$E_{\lim} = a_{Elim}\left(\exp(b_{Elim}(w_g - w_{resid})^2) - 1\right) \tag{A20}$$

$a_{Elim}$ and $b_{Elim}$ are model parameters related to soil evaporation response.

$R^{'}$ factor in Eq.(A13) accounts for stomatal resistance and to the fact that only the fraction of the canopy area which is not

covered by water will contribute to evapotranspiration. Deardorff (1978) proposed the expression:

$$R^{'} = \left(\frac{dew}{d_{\max}}\right)^{2/3} + \left[1 - \left(\frac{dew}{d_{\max}}\right)^{2/3}\right]\frac{1}{(\beta + C_{fh}RST)}, \tag{A21}$$

$$R^{'} = 1 \qquad \text{for condensation,}$$

where "dew" (resp. "$d_{max}$") is the fraction (resp. the maximal one) of free water on the foliage. RST is the stomatal resistance, this factor governs the canopy participation to the energy budget and is responsible for partition between sensible and latent

heat fluxes.

In the model, calculation of RST is based on Collatz et al. (1991, 1992) and is the same as in SiB models (Sellers et al., 1992, 1996a). Biophysical and environmental variables manage photosynthesis processes giving $CO_2$ assimilation rate and then stomatal conductance of the foliage.

Ball (1988) gives the following leaf stomatal conductance expression:

$$g_s = m\frac{A_n}{c_s}h_s p + b \tag{A22}$$

where $A_n$ is net assimilation rate calculated by the model of Farquhar et al. (1980), $c_s$ and $h_s$ are CO$_2$ partial pressure and relative humidity at leaf surface, $p$ is atmospheric pressure, $m$ and $b$ are empirical factors from observations depending on vegetation type (C$_3$ or C$_4$).

Assimilation rate is determined by means of three factors, a photosynthetic enzyme (Rubisco) limiting rate, a light limiting

rate and a limiting rate owing to the leaf capacity to export or utilize the photosynthesis products (Collatz et al., 1991). In the model, the iterative solution method for the photosynthesis-stomatal conductance calculation proposed by Collatz et al. (1991) has been implemented. Indeed, canopy is considered as a "big leaf" assuming bulk or integral values over canopy depth used in the integrated form of Eq.(A22) (see Sellers et al., 1992). Stomatal conductance and net assimilation rate are then determined for the canopy.

*Competing interests.* The authors declare that they have no conflict of interest.



*Acknowledgements.* The authors wish to acknowledge the French Ministry of Research for funding the PhD scholarship of Guillaume Bigeard as well as USGS, NASA and JPL for providing LANDSAT and ASTER thermal Infrared data. This work was also partly supported by the French PNTS and TOSCA programs (with the "Assimilation multicritère dans la modélisation TSVA: complémentarité des grands domains spectraux" PNTS project by J. Demarty, the "Validité des estimations d'évapotranspiration basées sur l'utilisation de don-

5 nées infrarouges thermiques" PNTS project by D. Courault and the EVA2-IRT TOSCA project by G. Boulet and A. Olioso), the Seventh Framework Programme (FP7) with the SIRIUS project, the MISTRALS ENVIMED and SICMED programs and the Joint International Laboratory TREMA (http://trema.ucam.ac.ma) which allows collaborating with UCAM University for the application of the models on the Morocco site. Finally, the authors would like to thank M. Anderson and W. Kustas from USDA-ARS for the dissemination of the TSEB model through the community which allows such research.





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



**Table 3.** Sites characteristics and overview of available cultures.

| Site | | Auradé | Lamasquère | Sidi Rahal |
|---|---|---|---|---|
| **Location** | | France | France | Morocco |
| **Latitude** | | 43.54984444 °N | 43.49737222 °N | 31.665852 °N |
| **Longitude** | | 1.10563611 °E | 1.23721944 °E | 7.597873 °W |
| **Climate** | | temperate | temperate | semi-arid |
| **Soil type** | | Clay loam | Clay | Clay |
| **sand[%] silt[%] clay[%]** | | 21 47 32 | 12 34 54 | 20 34 46 |
| **Depth [m]** | | 0.6 | 1 | 1 |
| **Slope [%]** | | 2 | 0 | 1 |
| **2004** | Culture | - | - | **Wheat *** |
| | Growth cycle length [$days$] | - | - | 133 |
| | Maximum $LAI$ [$m^2\ m^{-2}$] | - | - | 3.76 |
| | Cumulated rain [$mm$] | - | - | 135 |
| | Cumulated irrigation [$mm$] | - | - | 120 |
| **2006** | Culture | **Wheat** | **Corn *** | - |
| | Growth cycle length [$days$] | 246 | 123 | - |
| | Maximum $LAI$ [$m^2\ m^{-2}$] | 3.13 | 3.33 | - |
| | Cumulated rain [$mm$] | 397 | 132 | - |
| | Cumulated irrigation [$mm$] | 0 | 148 | - |
| **2007** | Culture | **Sunflower** | **Wheat** | - |
| | Growth cycle length [$days$] | 157 | 271 | - |
| | Maximum $LAI$ [$m^2\ m^{-2}$] | 1.74 | 4.47 | - |
| | Cumulated rain [$mm$] | 456 | 531 | - |
| | Cumulated irrigation [$mm$] | 0 | 0 | - |
| **2008** | Culture | **Wheat** | **Corn *** | - |
| | Growth cycle length [$days$] | 248 | 175 | - |
| | Maximum $LAI$ [$m^2\ m^{-2}$] | 2.39 | 3.28 | - |
| | Cumulated rain [$mm$] | 491 | 397 | - |
| | Cumulated irrigation [$mm$] | 0 | 50 | - |

* irrigated cultures.





**Table 4.** Intercomparison of TSEB and SEtHyS performances (RMSE), with influence of time resolution, phenological stage, culture and climate.

| | | RMSE [$W\ m^{-2}$] | | | | | |
| --- | --- | --- | --- | --- | --- | --- | --- |
| | | $R_n$ | | $H$ | | $LE$ | |
| | | TSEB | SEtHyS | TSEB | SEtHyS | TSEB | SEtHyS |
| **Time resolution** | Overall (time step) | 46.5 | 25.7 | 28.9 | 38.0 | 54.7 | 47.1 |
| | Overall (daily average) | 42.7 | 18.9 | 21.2 | 28.7 | 38.9 | 35.5 |
| **Phenology** | Rising | 22.1 | 15.3 | 110.2 | 44.1 | 88.3 | 44.0 |
| | Growth | 30.9 | 24.5 | 21.7 | 28.3 | 51.6 | 43.4 |
| | Max of vegetation | 51.1 | 20.2 | 24.6 | 40.8 | 55.5 | 48.1 |
| | Senescence | 55.0 | 29.4 | 43.5 | 47.3 | 54.0 | 42.1 |
| | Hydric stress | 53.2 | 21.6 | 44.9 | 49.3 | 49.6 | 30.6 |
| **Culture** | Wheat | 49.7 | 29.5 | 32.9 | 37.6 | 49.2 | 45.6 |
| | Corn | 46.0 | 18.1 | 22.9 | 40.2 | 64.4 | 52.6 |
| | Sunflower | 39.1 | 27.2 | 27.1 | 35.1 | 49.0 | 39.5 |
| **Climate** | France (wheat) | 35.1 | 32.6 | 35.1 | 36.4 | 52.5 | 42.9 |
| | Morocco (wheat) | 25.6 | 15.2 | 25.6 | 40.8 | 36.3 | 53.4 |



**Table 5.** Comparison of *in situ* data and spatial data (SAFRAN, ASTER, and inversed NDVI)

| Forcing | Source | Variables [unit] | Description | Mean error | | "Extreme" error | |
|---|---|---|---|---|---|---|---|
| | | | | RMSE | BIAS | $1^{st}$ decile | $9^{th}$ decile |
| **Meteo** | SAFRAN | $T_a$ [$°C$] | Air temperature | 1.5 | 0.7 | -1.5 (-10 %) | 1.3 (+10 %) |
| | | $U_a$ [$m\ s^{-1}$] | Wind speed | 1.4 | -0.7 | -0.65 (-30 %) | 2.3 (+90 %) |
| | | $R_h$ [%] | Relative humidity | 7 | 8 | -12 (-15 %) | 5(+8 %) |
| | | $R_g$ [$W\ m^{-2}$] | Global radiation | 90 | 35 | -186 (-40 %) | 125 (+60 %) |
| | | $R_a$ [$W\ m^{-2}$] | Atmospheric radiation | 30 | 14 | -51 (-15 %) | 20 (+7 %) |
| **Vegetation** | FORMOSAT | LAI [$m^2\ m^{-2}$] | Leaf Area Index | - | 20 % | -50 % | +50 % |
| | | $h_c$ [$m$] | Canopy height | - | 20 % | -100 % | +100 % |
| **LST** | ASTER | $T_s$ [$K$] | Surface temperature | 2 | - | - | - |