# Peer review of "Ability of a SVAT and a two-source energy balance model to predict evapotranspiration for several crops and climate conditions"

_Hydrology and Earth System Sciences, 2018_

## Referee Comment (RC1) · Anonymous Referee #1 · 17 Oct 2018

Overall Comments: The authors present a study that contrasts two distinct approaches to utilizing thermal infrared (TIR) remote sensing for estimating surface evapotranspiration: a surface energy balance model that is forced directly by TIR data, and a more complex SVAT model that assimilates TIR observations.

I believe this is an interesting topic to address, but I do not feel the authors have performed a satisfying comparison of these two methods. One source of major concern is the explanation of what exactly was done. For example, it remains unclear to me how TIR data is "assimilated" into the SVAT model. This needs to be very clearly described.

Likewise, the use of TIR for SEB appears to be through optimizing a few parameters, but this likewise remains unclear to me. It is also unclear exactly what years / seasons / crops were evaluated at each of the two sites (France and Morocco). At one point in the manuscript they mention two available meteorological stations, one in France and one in Morocco, which indicates that the data may be from multiple years. Was each crop evaluated for a single growing season at each site, multiple years for each site? The lack of clarity and detail about the methods used, particularly how TIR data is used to constrain, or for data assimilation, in each model makes it difficult to evaluate the results effectively.

The paper initially appears to be focused on the evaluation of two different types of models, surface energy balance versus a full SVAT model, for estimating evapotranspiration from thermal infrared remote sensing. At some point it transitions into a sensitivity analysis paper, which does not tie back to the original point as far as I could tell. A major revision should seek to bring out the use of TIR data in these two model frameworks, without a heavy focus on broad sensitivity analysis of the two models.

The English phrasing in the manuscript could generally be improved for greater clarity and to reduce confusion in some of the explanations of methods and results. Likewise, the use of a spell checker will catch a few spelling errors that exist in the reviewed manuscript.

Specific Comments:

1) Lines 10-11: The following statement in the Abstract should be re-phrased for clarity. "TSEB has been shown to be more flexible and requires one single set of parameters but lead to low performances on rising vegetation and stressed conditions. " It is not clear to me what "low performances on rising vegetation" means.

2) Lines 14-17: The final couple of sentences in the Abstract are confusing and should be re-written for clarity.

3) Section 2.2: While citations are provided for complete descriptions of each site, the first paragraph should include information on the explicit contrasts or similarities of the two sites, such as: where all three agricultural species grown at each of the two sites? How is irrigation managed / used at each site, and for each crop? What are the mean climate variables during the growing season such as temperature, VPD, precipitation?

4) Section 2.2: Clarify for what years / seasons each crop / site was monitored with meteorological instruments and eddy covariance. This should be clarified in the second paragraph of 2.2.

5) Section 2.3: Rather than using a few 10-day periods, why not use the full growing season records for each crop / site to more fully evaluate the capabilities of each model. I would think that the assimilation of TIR data into the SVAT would have a payoff that increases over time, as erroneous parameter values are further corrected / improved with each assimilation cycle.

6) Section 2.4: The authors mention that a "multi-objective calibration method" is used, with "five target functions". Please clarify what this means. What are the functions: a set of objective functions that each minimize the difference between a variable and the observed quantity? Or, are multiple objectives used here. The objective functions, and exactly what variables they pertain to, needs to be clarified.

7) Figure 2: The axis values for MAPD confuse me. In both cases they start at 43, decrease rapidly to 23, and then increase again to 53. I would expect monotonically increasing axis values...

8) Section 3.1: The authors previously state that water stress periods are primarily confined to the senescence phase, but here point out that the changes in canopy radiation transfer, pigment contents, etc, are not taken into account by TSEB. This goes to my earlier point that the entire set of growing seasons should be simulated and evaluated with both models, not just a few 10-day periods. Stress is likely to be found at both sites, either between irrigation events or rain events.

9) Figure 3: In the legend describe the difference between the top and bottom panels.

10) Section 3.2.3: It is hard to believe that the SVAT model is only sensitive to wind speed for LE computation, and not other meteorological inputs such as radiation forcing, or VPD. How do the authors explain this?

11) Figure 6: The relatively minor impact of biases in Ts (i.e. thermal infrared temperature measurements), relative to the reference RMSE, indicates that TSEB is not very sensitive to TIR inputs. Doesn't this contradict the premise that this is one of two models that can be used for ET monitoring from TIR data?

12) It would be very nice to see a figure identical to Figure 6, but for the SVAT model.

13) Section 3.26: At the end of this section the authors appear to argue that their two well-watered sites that do not apparently see significant water stress during the growing season may not be best suited to an experiment such as this, focused on evaluating two TIR-based ET approaches. I would tend to agree that at least an additional site that experiences significant periods of water stress throughout the growing season is merited.

14) Section 3.2.8: The authors state that the parameter Vmax0 has a different value at every time period for each crop. I don't understand what this means exactly. Is this parameter varied in the assimilation procedure, and it shows large variability from time period to time period? This data should be shown, even if in Supplementary.

---

## Referee Comment (RC2) · Anonymous Referee #2 · 18 Dec 2018

General comments:

The paper explores the application of the Two-Source Energy Balance (TSEB) using TIR data and a SVAT model (SEtHyS) over experimental agricultural sites in Morocco and France, hence different climate and management practices. With regards to application of TSEB, in the model description, it appears they are using most if not all of the original formulations of the Norman et al (1995) model, for example Eq. (9) for partitioning net radiation (Rn) for the soil and canopy elements. However later they state that they adopt a more physically-based Rn divergence model of Kustas and

Norman (1999). Yet in the sensitivity analysis (Table 1) this extinction coefficient for Eq. (9) is retained and evaluated later in Figure 9 which is not consistent with what is stated in the text. While in the references they appear to cite papers that have included new formulations being implemented in the TSEB since the original 1995 paper, they are not included in this paper. The TSEB model has undergone several modifications since it was first presented by Norman et al. (1995). Changes include refinements to the algorithm estimating soil aerodynamic resistance and shortwave and longwave transmittance through the canopy (as they mention in their paper; Kustas and Norman, 2000) and additionally a means for adjusting the Priestley–Taylor formulation for canopy transpiration (Kustas and Norman 1999). Further improvements include incorporating rigorous treatment of radiation modeling for strongly clumped row crops, accounting for shading effects on soil heat flux (Colaizzi et al. 2012a, 2016a,b), and incorporating alternative formulations for computing the canopy transpiration such as Penman–Monteith (PM) or light-use efficiency (LUE) parameterizations (see Colaizzi et al. 2012b, 2014,2016c; Anderson et al. 2008). The later two canopy transpiration formulations are mentioned but not applied in this paper. Alternatively, the SEtHyS is a SVAT model with 22 parameters and so it is unclear why such a comparison is actually being made between a relatively simple but fairly robust thermal-based model and a SVAT having a large number of tunable parameters. It's also unclear why this comparison does not include application of a newly developed and presumably more robust two-source model SPARSE developed by one of the co-authors (Boulet et al., 2015). Additionally, for the sensitivity analysis, the authors do not appear to be aware of the several studies that have already performed sensitivity analyses for key inputs to TSEB. These include two of the papers mentioned in this manuscript... Timmermans et al (2007) and Zhan et al. (1996). There is also Li et al (2005) mentioned in the manuscript and then there is the paper by Kustas and Norman (1997) and Kustas et al. (2012). In summary it appears they conduct an analysis with a dated TSEB model without some of the more current refinements and comparing it to a SVAT that has a number of tunable parameters that would be difficult to prescribe over a large area

without detailed ground information. There are a significant number of analyses performed making it a long paper and is somewhat diffuse in its focus. While I think the paper has some unique findings, it does not consider some of the main advances in TSEB when evaluating model performance for these agricultural sites. Early season conditions when the canopy is small, the soil is playing a major role in the energy exchange, and there is no discussion of soil roughness effects on the TSEB formulation that has been discussed in the literature (Kustas et al., 2016). Errors in TSEB during senescence will largely depend on how well the green fraction is determined. . .however it should be pointed out that these later stages of vegetation condition are not as important to capture the ET as during the main growing season. While I consider this work as having some merit, particularly the analyses performed with SEtHyS, it seems the authors do not consider to any degree of the advances/refinements made in the TSEB model since Norman et al (1995) and therefore I question how relevant is their analyses and conclusions using the 20+ year old formulations evaluated here in comparison to the more current parameterizations. Based on these shortcomings I do not find the paper suitable for publication in its current form.

Specific comments:

Page 9: It appears the leaf area and green fraction data are very local and may not reflect conditions viewed by the radiometer. This can be a major issue. Is there any indication where they sampled is representative of the radiometer field of view?

Page 9: Eq (15). What values are assumed in the Penman-Monteith equation for computing LEpot?

Page 10: How is the calibration of SEtHyS carried out and what level of calibration is shown in Figure 2 for the SEtHyS model?

Page 10: So the TSEB performance is "sought in its out-of-the box configuration presented in Norman et al (1995)" suggests none of the refinements over the last 20 years are incorporated in this analysis.

Page 10. The 3 parameters identified for study are the Priestley-Taylor coefficient, the net radiation extinction parameter and the fraction of soil net radiation for estimating soil heat flux, G. There is some interdependency here between the amount of canopy net radiation interception and the value of the Priestley-Taylor parameter (Kustas and Norman, 2000). Also for G, refinements of the TSEB include time varying formulation proposed by Santanello and Friedl (2003).

Page 12 line (10): TSEB could be provided albedo inputs from remote sensing. This is something easily done in the model if made available.

Page 12 (line 15): The authors do not seem to be aware of the soil resistance formulation that is sensitive to soil roughness which is discussed in refinements to the TSEB model (Kustas et al., 2016).

Page 12 (Line 30): Its unclear what version of SEtHyS model (1-4 from page 10) is being used in these comparisons.

Page 13 (line 5): The Crow et al (2008) paper actually showed the utility of TSEB in providing an indicator of plant stress for assimilation in a water balance model.

Page 15 Sensitivity analysis to meteorological inputs: It has been long recognized that to apply TSEB regionally requires a way of reducing the need for accurate absolute surface-air temperature differences. This was the motivation for the development of time differencing modeling schemes (Anderson et al., 1997; Norman et al., 2000).

Page 15 Sensitivity analysis to vegetation forcing inputs: The use of micrometeorological measurements close to the canopy height is ill-advised in general due to roughness sublayer effects and so comes as no surprise for the TSEB since the aerodynamic resistances are key to the TSEB calculations. This should be removed

Page 17: Sensitivity analysis to radiative temperature for TSEB: This is well documented and the reason why time differences in radiative temperatures were developed early in the TSEB applications (see Anderson et al., 2004)

[Figure]

Page 17-18: sensitivity analysis to water inputs and soil water content for SEtHyS: This is a major issue with SVAT models. That is why approaches like Crow et al (2008) of combining water balance with remote sensing energy balance is appealing. Moreover, for regional analysis it will be very difficult to acquire irrigation information in a timely manner.

Page 22 (figure 9): These results are related to some extent on the radiation partitioning which the authors appear to have adopted the original formulation of Norman et al (1995) for net radiation extinction and without any clumping effects which row crops tend to have (Anderson et al., 2005).

Page 25 (figure 11): Did the authors consider the fact that extinction of diffuse light through a canopy is quite different from direct and perhaps that is another factor affecting the Priestley-Taylor value?

References:

Anderson MC, Norman JM, Diak GR, Kustas WP, Mecikalski JR. (1997) A two-source time-integrated model for estimating surface fluxes using thermal infrared remote sensing. Remote Sens Environ 60:195–216.

Anderson, M.C., Norman, J.M., Mecikalski, J.R., Torn, R.D., Kustas, W.P., & Basara, J.B. (2004). A multi-scale remote sensing model for disaggregating regional flues to micrometeorological scales. J. Hydromet, 5, 343–363.

Anderson MC, Norman JM, Kustas WP, Li F, Prueger JH, Mecikalski JR (2005) Effects of vegetation clumping on two–source model estimates of surface energy fluxes from an agricultural landscape during SMACEX. J Hydromet 6(6):892–909

Anderson MC, Norman JM, Kustas WP, Houborg JM, Starks PJ, Agam N (2008) thermal-based remote sensing technique for routine mapping of land-surface carbon, water and energy fluxes from field to regional scales. Remote Sens Environ 112:4227–4241

Boulet, G., Mougenot, B., Lhomme, J.-P., Fanise, P., Lili-Chabaane, Z., Olioso, A., Bahir, M., Rivalland, V., Jarlan, L., Merlin, O., Coudert, B., Er-Raki, S., and Lagouarde, J.-P. (2015) The SPARSE model for the prediction of water stress and evapotranspiration components from thermal infra-red data and its evaluation over irrigated and rainfed wheat, Hydro Earth Sys. Sci. 19:4653–4672

Colaizzi PD, Evett SR, Howell TA, Li F, Kustas WP, Anderson MC (2012a) Radiation model for row crops: I. Geometric view factors and parameter optimization. Agron J 104:225–240

Colaizzi PD, Kustas WP, Anderson MC, Agam N, Tolk JA, Evett SR, Howell TA, Gowda PH, O'Shaughnessy SA (2012b) Two-source energy balance model estimates of evapotranspiration using component and composite surface temperatures. Adv Water Resour 50:134–151

Colaizzi PD, Agam N, Tolk JA, Evett SR, Howell TA, Gowda PH, O'Shaughnessy SA, Kustas WP, Anderson MC (2014) Two-source energy balance model to calculate E, T, and ET: comparison of Priestley–Taylor and Penman–Monteith formulations and two time scaling methods. Trans ASABE 57(2):479–498

Colaizzi PD, Evett SR, Agam N, Schwartz RC, Kustas WP (2016a) Soil heat flux calculation for sunlit and shaded surfaces under row crops: 1. Model development and sensitivity analysis. Agric For Meteorol 216:115–128

Colaizzi PD, Evett SR, Agam N, Schwartz RC, Kustas WP, Cosh MH, McKee LG (2016b) Soil heat flux calculation for sunlit and shaded surfaces under row crops: 2. Model test. Agric For Meteorol 216:129–140

Colaizzi PD, Agam N, Tolk JA, Evett SR, Howell TA, O'Shaughnessy SA, Gowda PH, Kustas WP, Anderson MC (2016c) Advances in a two-source energy balance model: partitioning of evaporation and transpiration for cotton using component and composite surface temperatures. Trans ASABE 59(1):181–197. https ://doi.org/10.13031 /trans

.59.11215

Crow, W. T., Kustas, W. P., and Prueger, J. H. (2008) Monitoring root-zone soil moisture through the assimilation of a thermal remote sensing-based soil moisture proxy into a water balance model, Remote Sens. Environ. 112:1268–1281.

Kustas WP, Norman JM. (1997) A two-source approach for estimating turbulent fluxes using multiple angle thermal infrared observations. Water Resour Res 33:1495–1508.

Kustas WP, Norman JM (1999) Evaluation of soil and vegetation heat flux predictions using a simple two-source model with radiometric temperatures for partial canopy cover. Agric For Meteorol 94:13–29

Kustas W, Norman JM (2000) A two-source energy balance approach using directional radiometric temperature observations for sparse canopy covered surfaces. Agron J 92(5):847–854

Kustas WP, Alfieri JG, Anderson MC, Colaizzi PD, Prueger JH, Evett SR, Neale CMU, French AN, Hipps LE, Chávez JL, Copeland KS, Howell TA. (2012) Application of a time differencing technique using local thermal observations in a strongly advective irrigated agricultural area. Adv Water Resour 50:120–33.

Kustas WP, Nieto H, Morillas L, Anderson MC, Alfieri JG, Hipps LE, Villagarcía L, Domingo F, García M (2016) Revisiting the paper "using radiometric surface temperature for surface energy flux estimation in mediterranean drylands from a two-source perspective. Remote Sens Environ 184:645–653

Li F, Kustas WP, Prueger JH, Neale CMU, Jackson TJ. (2005) Utility of remote sensing based two-source energy balance model under low and high vegetation cover conditions. J Hydrometeorol 6:878–91.

Norman JM, Kustas WP, Prueger JH, Diak GR. (2000) Surface flux estimation using radiometric temperature: a dual temperature difference method to minimize measurement error. Water Resour Res 36:2263–74.

Santanello J Jr, Friedl M (2003) Diurnal covariation in soil heat flux and net radiation. J Appl Meteorol 42(6):851–862

Timmermans, WJ, Kustas WP, Anderson MC, French AN. (2007) An intercomparison of the surface energy balance algorithm for land (SEBAL) and the two-source energy balance (TSEB) modeling schemes. Remote Sens Environ 108:369–84.

Zhan, X., W. P. Kustas, and K. S. Humes, (1996) An intercomparison study on models of sensible heat flux over partial canopy surfaces with remotely sensed surface temperature. Remote Sens. Environ., 58, 242–256.

---

## Author Comment (AC1) · 6 Jun 2019

We first wish to thank the reviewer for his useful comments and corrections that we have, for most of them, taken into account. Significant rewriting has been necessary. The point by point responses are detailed below. We believe that the article has been considerably improved.

Note: two versions of the manuscript are provided, one with corrections and rewriting in response to reviewers highlighted in green, another one without colors.

[Figure]

Overall Comments:

[1] I believe this is an interesting topic to address, but I do not feel the authors have performed a satisfying comparison of these two methods. One source of major concern is the explanation of what exactly was done. For example, it remains unclear to me how TIR data is "assimilated" into the SVAT model. This needs to be very clearly described. Likewise, the use of TIR for SEB appears to be through optimizing a few parameters, but this likewise remains unclear to me.

We agree completely concerning the TIR data. We aimed to place this specific study within a larger context where TIR data are intended to be used to constraint the SVAT model trajectories through data assimilation. Within this study, data assimilation of TIR data within the SVAT model has not been implemented as we believe that a preliminary analysis of model calibration and sensitivity study to input errors is necessary. To this objective, a Multiobjective Calibration Iterative Procedure (MCIP) has been implemented to tune its parameters in the view of using TIR data. As this will be the subject of further work, we don't mention the use of TIR data into the SVAT model anymore in the abstract and in several places of the new version of the manuscript, following the referee comments.

We have modified the title of the paper to clarify the purpose of our study.

[2] It is also unclear exactly what years / seasons / crops were evaluated at each of the two sites (France and Morocco). At one point in the manuscript they mention two available meteorological stations, one in France and one in Morocco, which indicates that the data may be from multiple years. Was each crop evaluated for a single growing season at each site, multiple years for each site?

Concerning meteorological data, two stations are used, one in France and one in Morocco. Concerning micro-meteorological data (including latent and sensible heat fluxes), 3 fields have been instrumented: two in France (Auradé and Lamasquère sites) and one in Morocco (Sidi Rahal site). During the 3 years of study in France, crop rotations allowed us to gather data on wheat (3 seasons), maize (2 seasons) and sunflower (1 season). In Morocco, we used 1 crop season. We believe that table 1 clearly shows the number of growth season for each crop but we have also reformulated section 2.2 in the new version of the manuscript.

[3] The lack of clarity and detail about the methods used, particularly how TIR data is used to constrain, or for data assimilation, in each model makes it difficult to evaluate the results effectively.

Agree. Cf. point [1].

[4] The paper initially appears to be focused on the evaluation of two different types of models, surface energy balance versus a full SVAT model, for estimating evapo-transpiration from thermal infrared remote sensing. At some point it transitions into a sensitivity analysis paper, which does not tie back to the original point as far as I could tell. A major revision should seek to bring out the use of TIR data in these two model frameworks, without a heavy focus on broad sensitivity analysis of the two models.

We agree that the initial version of the manuscript was confusing. To our opinion, the different modeling frameworks of the two approaches, mainly the solving of a soil hydric budget for SETHYS and the use of surface temperature as an indirect proxy of the crop hydric conditions for TSEB, deserve a sensitivity analysis. The abstract has been rewritten (cf. point [1]) to point out that the paper is mainly dedicated to a sensitivity analysis of the two approaches based on a unique database. With regards to the use of TIR data, the cross sensitivity analyses of the models through the linkage of the radiative temperature and the SWC shows the different response of the models to the crop hydric conditions. These results can then be analyzed in the light of the models performance from the sensitivity analysis. Following the reviewer's comment and in order to match the objectives presented in the abstract and the content of the work, the abstract and the introduction have been reformulated. See also response to point [1].

[5] The English phrasing in the manuscript could generally be improved for greater clarity and to reduce confusion in some of the explanations of methods and results. Likewise, the use of a spell checker will catch a few spelling errors that exist in the reviewed manuscript.

OK. The manuscript has been reviewed by a native english speaker.

Specific Comments:

1) Lines 10-11: The following statement in the Abstract should be rephrased for clarity. "TSEB has been shown to be more flexible and requires one single set of parameters but lead to low performances on rising vegetation and stressed conditions. " It is not clear to me what "low performances on rising vegetation" means.

Agree. In the new version of the manuscript, this sentence has been replaced by : "TSEB is run with only one set of parameters and provides acceptable performances for all crop stages apart from the early beginning of the growing season (LAI < 0.2 m2.m−2 ) and when water stress or senescence occurred."

2) Lines 14-17: The final couple of sentences in the Abstract are confusing and should be rewritten for clarity.

Agree. The corresponding sentences of the abstract have been rewritten.

3) Section 2.2: While citations are provided for complete descriptions of each site, the first paragraph should include information on the explicit contrasts or similarities of the two sites, such as: where all three agricultural species grown at each of the two sites? How is irrigation managed / used at each site, and for each crop? What are the mean climate variables during the growing season such as temperature, VPD, precipitation?

Cf point [2] of the overall comments.

4) Section 2.2: Clarify for what years / seasons each crop / site was monitored with meteorological instruments and eddy covariance. This should be clarified in the second

paragraph of 2.2.

Cf point [2] of the overall comments.

5) Section 2.3: Rather than using a few 10-day periods, why not use the full growing season records for each crop / site to more fully evaluate the capabilities of each model. I would think that the assimilation of TIR data into the SVAT would have a payoff that increases over time, as erroneous parameter values are further corrected / improved with each assimilation cycle.

OK. TIR data is not assimilated into the SVAT model (cf point [1]). By contrast, sets of parameters representing specific phenological stages and hydric conditions (stressed/unstressed) are sought in the view of a future application of the SVAT model at the scale of a heterogeneous agricultural landscape. Consequently, the periods should be long enough to gather a sufficient amount of data (of good quality, meaning a good closure for the energy budget) and not too long so that the crop and hydric conditions don't change too much. 10-days has been shown in several studies to be a good tradeoff. Section 2.3 was reformulated in the new version of the manuscript to stress this aspect.

6) Section 2.4: The authors mention that a "multi-objective calibration method" is used, with "five target functions". Please clarify what this means. What are the functions: a set of objective functions that each minimize the difference between a variable and the observed quantity? Or, are multiple objectives used here. The objective functions, and exactly what variables they pertain to, needs to be clarified.

OK. Five objective functions are optimized simultaneously. The five objective functions are detailed in section 2.4 (l.7, p.11). They are built to minimize the distance between model predictions and observations thanks to the Root Mean Square Error (RMSE). An ensemble of simulations based on a monte-carlo sampling of the parameter space is carried out. For each simulation corresponding to a specific parameter set, five objective functions are computed (RMSE of LE, H, Rn, Tb, W_rz). The joint optimization

of these 5 objective functions is obtained following a Pareto ranking. Basically, a simulation is classified as "better" than the others if all the objective functions have lower values. For more details the MCIP methodology is described in Demarty et. al, 2004 and 2005.

7) Figure 2: The axis values for MAPD confuse me. In both cases they start at 43, decrease rapidly to 23, and then increase again to 53. I would expect monotonically increasing axis values. . .

Agree. It seems that exportation to pdf went wrong, axis values have been corrected in the new version of the manuscript.

8) Section 3.1: The authors previously stated that water stress periods are primarily confined to the senescence phase, but here point out that the changes in canopy radiation transfer, pigment contents, etc, are not taken into account by TSEB. This goes to my earlier point that the entire set of growing seasons should be simulated and evaluated with both models, not just a few 10-day periods. Stress is likely to be found at both sites, either between irrigation events or rain events.

Thank you for your comment. We agree that this point deserves clarification. It appears that irrigation was properly scheduled for all our study sites and seasons. Consequently, stress didn't occur during the growth phase of the crops. This has been carefully checked by looking at the whole time series (and not on 10-days period) of the measured root-zone soil moisture (through the SWI, cf. section 2.3) and the ratio between potential and real evapotranspiration (SE indicator, section 2.3). Stated differently, the SWI and SE indicators have been computed for the entire crop seasons and indeed stress periods are limited in time and occur basically during senescence at our study sites.

With regards to the 10-days periods, experimental data are uncertain by nature, subject to acquisition problems and, specifically for eddy-covariance systems, the energy balance closure is not guaranteed. By working on 10-days periods, we are sure that

the set of observations is complete and, for eddy covariance measurements, that the energy balance closure is good (>80% as stated in the new version of the manuscript in section 2.3, p.9, l.30).

For both models, SEtHyS and TSEB, the changes in canopy radiation transfer are taken into account by changes of the fraction of green (based on the LAI which is a model input) and the soil and vegetation albedos which are tunable parameters. As a consequence, the senescence phase is treated equally and with equal capabilities as other periods from the radiative transfer point of view.

9) Figure 3: In the legend describe the difference between the top and bottom panels.

Thank you. Legend was reformulated for more clarity.

10) Section 3.2.3: It is hard to believe that the SVAT model is only sensitive to wind speed for LE computation, and not other meteorological inputs such as radiation forcing, or VPD. How do the authors explain this?

OK. The SVAT model solves the surface energy balance. The radiative forcing in the short and long wavelengths is obviously one of the main drivers of the convective fluxes as shown in figure 4. Likewise, VPD, even if it is not a direct input of the model, significantly impact LE flux predictions following the formulation of LE based on a gradient of vapor pressure. Nevertheless, in this study, we considered typical errors that may be expected on input forcing when scaling up to the agricultural landscape. At this scale, some input variables will be more uncertain than another. For instance, at the landscape scale, a measure or an estimation of Rg can be obtained accurately while wind speed, that may be derived from large-scale re-analysis, is always very uncertain as it depends on local conditions. This is the reason why wind speed appears more impacting than radiative forcing in our study. Moreover, it is important to note that a white noise is added to the meteorological forcing and that the RMSE for reference simulations on H and LE are equal or superior to 30 W.m-2 and 50 W.m-2. As a consequence, the addition of a white noise can bring some compensation with this level of

reference error. Following the reviewer's comment, the text of section 3.2.3 has been updated.

11) Figure 6: The relatively minor impact of biases in Ts (i.e. thermal infrared temperature measurements), relative to the reference RMSE, indicates that TSEB is not very sensitive to TIR inputs. Doesn't this contradict the premise that this is one of two models that can be used for ET monitoring from TIR data? OK. We got the reviewer point. Nevertheless, surface temperature is only a proxy of hydric conditions when water is limiting. As most of our study sites are irrigated and located in temperate areas (at least for the french sites), water limiting conditions only occurs during short periods. It is likely that when focusing on water stressed periods, the sensitivity to surface temperature error for TSEB is expected to be higher. Nevertheless, as stated in point [8], stress periods are almost absent from our data set during the growth period. Finally, it has been shown that the assumption of a canopy transpiring at a potential rate in this model (even it is can be bypassed in some conditions) is strong and limit the sensitivity of the model to surface temperature errors.

12) It would be very nice to see a figure identical to Figure 6, but for the SVAT model.

As explained in point [1], TIR data are not assimilated in the SVAT model but used to constrain the SVAT model multiobjective calibration (MCIP methodology). The specific contribution of TIR data to SEtHyS model calibration was published in Coudert et al. 2006, 2007 and 2008. An equivalent to the Figure 6 can be found in Coudert and Ottlé 2007 (Figure 3). We agree that this was not clearly stated in the previous version of the manuscript. Several rewritings all over the manuscript have been proposed to avoid the confusion.

13) Section 3.26: At the end of this section the authors appear to argue that their two well-watered sites that do not apparently see significant water stress during the growing season may not be best suited to an experiment such as this, focused on evaluating two TIR-based ET approaches. I would tend to agree that at least an additional site

that experiences significant periods of water stress throughout the growing season is merited.

Yes. This is a very relevant issue and we agree that water stressed period are very occasional within our database apart from the end of the growing period when vegetation is senescent. That is the reason why we recently conducted a new experiment in Morocco (seasons 2017-2018 and 2018-2019) focused on water stress during which stress was intentionally triggered on one wheat field. Nevertheless, the processing and use of these new data is beyond the scope of the paper. These limits have been stressed again in the new version of the manuscript when the performances of the models during stress periods are analyzed and also in the conclusion part.

14) Section 3.2.8: The authors state that the parameter Vmax0 has a different value at every time period for each crop. I don't understand what this means exactly. Is this parameter varied in the assimilation procedure, and it shows large variability from time period to time period? This data should be shown, even if in Supplementary.

We are sorry that it was not clear in the former version of the manuscript, but indeed what was called "assimilation procedure" in the previous version of the manuscript, referred to adjustment/optimization of the parameters values. It has been changed to "calibration procedure" in the new version. In addition, among the 22 parameters, Vmax0, which represents the leaf photosynthesis capacity of Rubisco (Table 2), and which affects the assimilation rate and consequently the global evapotranspiration flux, was identified as one of the most sensitive. From a period to another, the calibration procedure leads to a variability of the optimal values of the Vmax0 parameter along the growth season. The calibration results show generally higher values during the periods with higher vegetation LAI where the evapotranspiration flux is maximal. Another coherent result (according to measurement of photosynthesis assimilation rate and stomatal conductance) is the lower values of Vmax0 obtained for corn and sunflower than for wheat. The figure 1 (attached to our answers) illustrates these results.

However, in order to clarify and focus the paper on its main objectives, we have removed the discussion on the parameters sensitivity analysis and calibration (section 3.2.8 in the former version of the manuscript).

[Figure]
Interactive
comment

[Figure]

**Fig. 1.**

---

## Author Comment (AC2) · 6 Jun 2019

We first wish to thank the reviewer for his useful comments and corrections that we have, for most of them, taken into account. Significant rewriting and some new analyses have been necessary. The Point by point responses are detailed below. We believe that the article has been considerably improved.

Note: two versions of the manuscript are provided, one with corrections and rewriting in response to reviewers highlighted in green, another one without colors.

[Figure]

General comments:

[1] With regards to application of TSEB, in the model description, it appears they are using most if not all of the original formulations of the Norman et al (1995) model, for example Eq. (9) for partitioning net radiation (Rn) for the soil and canopy elements. However later they state that they adopt a more physically-based Rn divergence model of Kustas and Norman (1999). Yet in the sensitivity analysis (Table 1) this extinction coefficient for Eq. (9) is retained and evaluated later in Figure 9 which is not consistent with what is stated in the text.

Agree. The word "out-of-the box" used in the introduction was certainly confusing as we wanted to state that we had implemented and tested most of the improvements published since the original version of TSEB, apart from the Penman-Monteith version. The short answer is that the added value of most of the tested improvement were not clear with our database. This is why we choose the term "out-of-the-box" and also for the simplicity of the presentation in the previous version of the manuscript. The detailed responses are listed at point [2]. The introduction has been reformulated (also in response to reviewer 1) and "out-of-the-box" has been removed.

Concerning the specific formulation of net radiation, we agree there has been a mismatch in the text as the formulation used in our study is the one presented Anderson et al. (1997), following the Beer's extinction law and accounting for the dependence to the solar zenith angle (Rnsoil=Rn*exp(-Kapa LAI/sqrt(2.cos(phi)))). We've also tested the new formulation for clumped crops as clumped crops may intercept a lower part of the incoming radiation than if leaves were randomly distributed (RnSoil=Rn * exp(-Kapa*Gamma*LAI)) as introduced by Campbell (1998). Following Kustas et al., AFM (1999), we also tested the formulation of radiative budget separating short and long wavelengths. However the latter two formulation didn't improve the estimations of LE with our dataset and were not retained. We give more detail about this point in the next question. This has been corrected in the new version of the manuscript and the reference to the improvement with regard to the initial Norman et al. (1995) formulation
are properly referenced in the text following the reviewer (new section 2.4, p.11, l.24).

[2] While in the references they appear to cite papers that have included new formulations being implemented in the TSEB since the original 1995 paper, they are not included in this paper. The TSEB model has undergone several modifications since it was first presented by Norman et al. (1995).

As stated above, we had already tested several of the new development listed by the reviewer. The detailed responses are given in table 1 (attached to our answers). Following these results, we choose to keep the following parameterization in our version of the TSEB model: - Rn: Anderson, 1997 - G: Norman et al., 1995 - Transpiration: Anderson, 1997 (Priestley-Taylor) - Surface resistance: Norman et al., 1995 - Aerodynamic resistance: Norman et al., 1995

* refinements to the algorithm estimating soil aerodynamic resistance and shortwave and longwave transmittance through the canopy (as they mention in their paper; Kustas and Norman, 2000)

We agree with the reviewer that our text was (very) confusing. Cf. point [1].

In addition, considering the specific point of the formulation of net radiation, we agree that there has been a mismatch in the text. Both the Kustas and Norman (2000) and a modified version of the formulation of Norman et al. (1995) including the gamma factor for clumped canopies have been tested and obtained results were very close: the Root Mean Square Error (RMSE) of net radiation were 46.5 W/m$^2$ for the version derived from Norman et al. (1995) and 61.5 W/m$^2$ for the version of Kustas and Norman (1999). The modified original formulation providing with slightly better results, we choose to keep it in our study. Concerning the soil resistance, the version of Kustas and Norman (2000) was also adopted in our version of TSEB (cf. their equations 7 and 8). The presentation of the TSEB model has been reformulated and we hope that the parameterizations we used are now clearly stated.

[Figure]

* a means for adjusting the Priestley–Taylor formulation for canopy transpiration (Kustas and Norman 1999)

OK. Actually, the adjustment of the PT coefficient was also already activated in our version of TSEB. This is now clearly stated in the new version of the manuscript (section 2.1.2, p.5, l.24). Nevertheless, only marginal improvement are obtained as 0.7% of unlikely soil condensation cases occurs in our database at the half hourly time step.

* incorporating rigorous treatment of radiation modeling for strongly clumped row crops, accounting for shading effects on soil heat flux (Colaizzi et al. 2012a, 2016a,b)

OK. The Kustas and Norman, Agr. Jour., (2000) and Kustas and Norman, AFM (1999) modifications for sparse canopies and clumped crops were not adopted (cf. point [1] and point [2] above) as they do not give better performances on our crops and surface conditions (see results in the table 1 attached to our answers). The results of the sensitivity analysis to the Kappa parameter showed that optimal values were close to the original Norman et al. (1995) value of 0,45 and a clumping factor Gamma equal to 1. The only exception was sunflower, whose optimal value was about 0,75 (see Fig.9a of our former version of manuscript) corresponding to a clumping factor above 1 suggesting a clumped row. This seems consistent with the geometry of sunflower. The last developments of Colaizzi 2012a were not implemented because no separate measurements of bare soil and canopy temperatures were available in our database, as a consequence no validation was conceivable. However, we thank you for this comment because a recent experimental campaign in a small watershed (Auradé site of CESBIO in the south west france) with surface radiometric temperature measurements and "thermodynamical" soil and vegetation temperatures measurements with thermo-buttons has been carried out and will be used in further work to investigate this point.

* incorporating alternative formulations for computing the canopy transpiration such as Penman–Monteith (PM) or light-use efficiency (LUE) parameterizations (see Colaizzi

et al. 2012b, 2014, 2016c; Anderson et al. 2008). The later two canopy transpiration formulations are mentioned but not applied in this paper.

Agree. To our opinion, the incorporation of new formulations of canopy transpiration was an important drawback of our study. The Penman-Monteith (PM) approach has thus been implemented and tested. The comparison results between the PM and the Priestley-Taylor (PT) approaches are summarized in the table 2 (attached to our answers) for all data (for the half-hourly time step and for daily average) and then by stages of growth and by crops.

LE dominates the partition of convective fluxes within our irrigated sites and percentage of errors may thus be high on sensible fluxes as they exhibit much lower absolute values than LE fluxes. The two versions are thus quite close in terms of fluxes predictions, in particular for LE but the PT version is systematically better. A deeper look at the results shows that LE is strongly overestimated by the PM version, mainly during the rising and the growth stages of growth. This leads to significantly higher RMSE and MAPD while correlation coefficients remain close between the two versions. This could probably be related to the fact that our sites are located in relatively wet environment (the moroccan site is located at the center of an irrigated perimeter of 3000 ha while the sites in south western France are also surrounded by crop fields). The introduction in the parameterization of LE of a dependance to wind speed aiming to better represent advection fluxes in the PM version doesn't achieve the expected improvement within this specific conditions.

With regards to the results presented above, we choose not to retain the PM formulation of transpiration.

[3] Alternatively, the SEtHyS is a SVAT model with 22 parameters and so it is unclear why such a comparison is actually being made between a relatively simple but fairly robust thermal-based model and a SVAT having a large number of tunable parameters.

OK. The approaches are compared as they are both extensively used to map evapotranspiration and because modeling concepts are fundamentally different: for SEB models, surface temperature is the proxy for the crop hydric conditions while the hydric conditions are predicted thanks to a mechanistic water budget for SVATs. Several teams including our are working on the joint use of both approaches through the assimilation of snapshot evapotranspiration estimates to constrain the SVAT "continuous" predictions following Crow et al. (2005). To this objective, errors and biases of both modeling frameworks must be characterized carefully with regards to phenological stages and input data. To our knowledge, this is done in this paper thanks to a unique database as it includes several sites and several seasons. The introduction and the abstract have been reformulated following the reviewer comment in the view to better described the objectives, to highlight that the comparison is carried out on a large and unique data base in the sense that the cropping conditions of our study sites are quite specific.

[4] It's also unclear why this comparison does not include application of a newly developed and presumably more robust two-source model SPARSE developed by one of the co-authors (Boulet et al.,2015).

When the work has been carried out, SPARSE model was under development. Nevertheless, recent comparison between the TSEB model and SPARSE have shown that SPARSE was very close to TSEB (Boulet et al., 2015). The comparison to SPARSE is beyond the scope of the paper but SPARSE will probably be adopted in further studies.

[5] Additionally, for the sensitivity analysis, the authors do not appear to be aware of the several studies that have already performed sensitivity analyses for key inputs to TSEB. These include two of the papers mentioned in this manuscript. . . Timmermans et al (2007) and Zhan et al. (1996). There is also Li et al (2005) mentioned in the manuscript and then there is the paper by Kustas and Norman (1997) and Kustas et al. (2012).

OK. Thank you for the references. Zhan et al., 1996 intercompared model for sensible

heat fluxes. The sensitivity analysis was also carried out with regards to H only. In addition, forcing meteorological variables such as incoming radiation and air humidity were not considered in this study. Finally, the database comprises a much lower number of observations than ours. Li et al. (2005) is focused on the sensitivity to Leaf Area Index (and thus fractional cover Fc as an empirical relationship relating Fc to LAI is used). They highlighted that introducing a 20% bias in LAI lead to about 15-20% difference in H for the parallel version of TSEB and about half of these values for the series version (5-10%) demonstrating the higher robustness of this latter version with regards to LAI/Fc inputs. Kustas and Norman (1997) analyzed the sensitivity of four versions of the two source model: the initial version of Norman in its parallel and series configuration and two version taking advantage of two view angle of Tb. The selected input variables for the sensitivity analysis are: wind speed (bias of 50%), air temperature (bias of 3K), LAI (bias of 50%), green fraction (reduction of 0.2) and radiometric temperature (bias of 1.5K). Kustas et al. (2012) performed a "worst case scenario" on one specific day of acquisition adding what they called "large" errors on Ta (+1 and +3K), Tr (-2K) and wind speed (+1.5 m/s) while biases appeared quite similar to the previous studies. The study of Timmermans et al. (2007) is quite similar as a bias of 25% is added or subtracted to Tr, Ta, u, z0m, LAI, fc, vegetation height hc. One specific day is chosen.

With regards to the sensitivity analysis presented above, our study is positioned quite differently as (1) realistic errors (errors that can be expected when applying the models over a heterogeneous agricultural landscape) are applied including both white noise and biases and (2) we cover larger growth, crops and hydric conditions thanks to our unique database.

[6] In summary it appears they conduct an analysis with a dated TSEB model without some of the more current refinements and comparing it to a SVAT that has a number of tunable parameters that would be difficult to prescribe over a large area without detailed ground information. There are a significant number of analyses performed

making it a long paper and is somewhat diffuse in its focus. While I think the paper has some unique findings, it does not consider some of the main advances in TSEB when evaluating model performance for these agricultural sites.

Agree. Cf point [1] and [2] with regards to the considered refinements and point [3] together with responses to reviewer 1 regarding the focus of the paper. We believe that the significant rewriting of the manuscript together with the new analysis that have been carried out make the paper clearer now.

[7] Early season conditions when the canopy is small, the soil is playing a major role in the energy exchange, and there is no discussion of soil roughness effects on the TSEB formulation that has been discussed in the literature (Kustas et al., 2016).

Agree. A mention to the need for a potential adaptation of the soil resistance parameterization for rough and partially vegetated surfaces, very likely conditions at the beginning of the crop season and after harvest, is added at (section 3.1, p.13, l.22) referencing Kustas et al. (2016).

[8] Errors in TSEB during senescence will largely depend on how well the green fraction is determined...However it should be pointed out that these later stages of vegetation condition are not as important to capture the ET as during the main growing season.

Agree. This was added in the manuscript (section 3.1, p.13, l.15).

[9] While I consider this work as having some merit, particularly the analyses performed with SEtHyS, it seems the authors do not consider to any degree of the advances/refinements made in the TSEB model since Norman et al (1995) and therefore I question how relevant is their analyses and conclusions using the 20+ year old formulations evaluated here in comparison to the more current parameterizations. Agree. Cf. point[1] and point[2].

Specific comments:

Page 9: It appears the leaf area and green fraction data are very local and may not
reflect conditions viewed by the radiometer. This can be a major issue. Is there any indication where they sampled is representative of the radiometer field of view?

This is indeed a potential major issue. On the experimental sites, special attention is given to sample vegetation on areas representative of the radiometers' fields of view.

Page 9: Eq (15). What values are assumed in the Penman-Monteith equation for computing LEpot?

OK. Potential evaporation was computed with the classical Penman-Monteith formulation (and not ET0 from the modified FAO56 approach). The jarvis formulation extrapolated to the canopy based on LAI is used for the canopy resistance and the minimum resistance is equal to 90 s/m. This has been specified in the new version of the manuscript (section 2.3, p.9, l.31).

Page 10: How is the calibration of SEtHyS carried out and what level of calibration is shown in Figure 2 for the SEtHyS model?

OK. More details about the MCIP methodology were added in the manuscript, in particular in response to reviewer 1. 5 objective functions are optimized simultaneously. The five objective functions are detailed in section 2.4. They are built to minimize the distance between model predictions and observations thanks to RMSE. An ensemble of simulations based on a monte-carlo sampling of the parameter space is carried out. At each simulation (based on a specific parameter set) corresponds several objectives functions to minimize (RMSE of LE, H, Rn, Tb, W_rz). The global minimization is obtained following a Pareto ranking based on these objectives functions. Basically, a simulation is classified as "better" than another (thus at a lower rank) if all these objectives functions have lower values. For more details the MCIP methodology is described in Demarty et. al, 2004 and 2005.

Figure 2 presents simulation with optimal set of parameters as stated in section 2.4. This was added in the caption for more clarity.

Page 10: So the TSEB performance is "sought in its out-of-the box configuration presented in Norman et al (1995)" suggests none of the refinements over the last 20 years are incorporated in this analysis.

Indeed, this was badly formulated. The manuscript was updated with better overview of improvements tested and used. Cf. point [1] and point [2].

Page 10. The 3 parameters identified for study are the Priestley-Taylor coefficient, the net radiation extinction parameter and the fraction of soil net radiation for estimating soil heat flux, G. There is some interdependency here between the amount of canopy net radiation interception and the value of the Priestley-Taylor parameter (Kustas and Norman, 2000). Also for G, refinements of the TSEB include time varying formulation proposed by Santanello and Friedl (2003).

Yes. Agree with the reviewer. Thank you. A sensitivity analysis carried out by Diarra et al. (2017) demonstrated these equifinality issues between Kapa and the Priestley-Taylor coefficient but this study also showed that the partition of available energy between H and LE is quite robust with regards to these parameters values.

Concerning other formulations of the conduction fluxes, the parameterization of Santanello and Friedl (2003) was also tested but didn't provide an improvement of the results (cf. table at point [2]).

Page 12 line (10): TSEB could be provided albedo inputs from remote sensing. This is something easily done in the model if made available.

Agree. Thank you. The objectives of the study was to evaluate the best performance from the best possible inputs. The observed albedo were thus used. As stated by the reviewer, accurate albedo can be obtained from remote sensing observations either by computing specific empirical equations to some band reflectances (usually red and NIR bands) or using directly products (such as MODIS).

Page 12 (line 15): The authors do not seem to be aware of the soil resistance formulation that is sensitive to soil roughness which is discussed in refinements to the TSEB model (Kustas et al., 2016).

We thank the reviewer for his suggestion. This paper shows that the values proposed for the a and b coefficients of the rs parameterization (Norman et al., 1995) are associated with an underestimation of latent heat fluxes for sparse vegetation in semi-arid conditions. Nevertheless, the absolute values of LE are low during the emerging and rising stages and even if high relative error values (MAPD) are highlighted, absolute values are limited.

Page 12 (Line 30): Its unclear what version of SEtHyS model (1-4 from page 10) is being used in these comparisons.

Agree. The performances and sensitivity analyses presented were done with the "optimal" sets of parameters, i.e. the sets of parameters processed for each phenological stage and culture class. This is stated clearly in the manuscript (section 2.4, p.10, l.23).

Page 13 (line 5): The Crow et al (2008) paper actually showed the utility of TSEB in providing an indicator of plant stress for assimilation in a water balance model.

This paper is very interesting and is a good illustration of the possible complementarity between TSEB and a WEB-SVAT. We hope that the focus of our study is now more precisely explained. We precisely intent to bring elements concerning the domains of validity of the models and their performances through a variety of surface and meteorological conditions, taking into account models parameters and inputs sensitivity in order to consider the different couplings between both approaches for agricultural landscape spatialization purposes.

Page 15 Sensitivity analysis to meteorological inputs: It has been long recognized that to apply TSEB regionally requires a way of reducing the need for accurate absolute surface-air temperature differences. This was the motivation for the development of time differencing modeling schemes (Anderson et al., 1997; Norman et al., 2000).

Yes exactly, the question of the coupling between air temperature and surface conditions is determining in the surface budget and the convective fluxes calculation. Our proposition is to compare local measurements to regional estimations of air temperature from reanalysis in order to have a realistic uncertainty on model input when simulation run at field scale or homogeneous entity scale for SEtHyS and pixel scale for TSEB for regional application. Time differences in surface temperature methodology for estimating relevant air temperature is suited to large scale applications while in our study we propose to estimate the uncertainty for field scale simulation for landscape/regional application. The text of section 3.2.1 has been modified to make this clearer.

Page 15 Sensitivity analysis to vegetation forcing inputs: The use of micrometeorological measurements close to the canopy height is ill-advised in general due to roughness sublayer effects and so comes as no surprise for the TSEB since the aerodynamic resistances are key to the TSEB calculations. This should be removed.

Yes, this joins our previous answer. TSEB is supposed to be applied at high resolution TIR pixel (i.e. Landsat 8).

Page 17: Sensitivity analysis to radiative temperature for TSEB: This is well documented and the reason why time differences in radiative temperatures were developed early in the TSEB applications (see Anderson et al., 2004)

Page 17-18: sensitivity analysis to water inputs and soil water content for SEtHyS: This is a major issue with SVAT models. That is why approaches like Crow et al (2008) of combining water balance with remote sensing energy balance is appealing. Moreover, for regional analysis it will be very difficult to acquire irrigation information in a timely manner.

Yes, indeed. The Crow et al. (2008) is one of the founding paper for our work. How to combine a complex SVAT that suffers from uncertain water inputs (irrigation) at the plot scale and an energy budget model providing snapshot evapotranspiration estimate from instantaneous surface temperature observations ? Characterizing the model errors and the domain of validity of both models is the prerequisite step that we develop in this study before a joint use of both approaches through data assimilation. We hope that the positioning and the objectives of our study are more clearly stated in the new version of the manuscript.

Page 22 (figure 9): These results are related to some extent on the radiation partitioning which the authors appear to have adopted the original formulation of Norman et al (1995) for net radiation extinction and without any clumping effects which row crops tend to have (Anderson et al., 2005).

OK. This has been clearly stated in the previous point, in particular point [1] and [2].

Page 25 (figure 11): Did the authors consider the fact that extinction of diffuse light through a canopy is quite different from direct and perhaps that is another factor affecting the Priestley-Taylor value?

We totally agree with the reviewer remark. Overcast and clear skies conditions are treated the same way in the calculation of the net radiation and of the radiation divergences through the canopy in the version of TSEB we used. However the more physically-based description described in Campbell et al. 1998 and implemented in Kustas & Norman 1999 with specific extinction for diffuse or direct radiation through the canopy was tested but did not give better results. Clear sky radiation is mainly direct while overcast radiation is more diffuse, which is certainly affecting the Priestley-Taylor coefficient value. A proposition to take this into account would be to use the "SKYL" factor (accounting for the ratio between sky irradiance and total (sun + sky) irradiance) instead of modulating the Priestley-Taylor coefficient according to the cloudiness. Our dataset doesn't include "SKYL" in situ measurements, it could be eventually estimated with some error. In our study, the "SKYL" is not taken into account neither for TSEB nor for SEtHyS and the physical description of the radiation divergence through the canopy are consistent between both models. The interest to propose directly a modulation of the Priestley-Taylor coefficient in such conditions lies in the multiplicity of the factors

that should be determined in this case of low atmospheric demand in term of evapo-transpiration simulation. In our sense, this goes in the direction of the conclusions of Kustas & Norman 1999. This comment was added in the manuscript in section 4.2 (p.24, l.5).
* * *
| Term | Formulation | Rn | H | LE | G |
|------|-------------|------|------|------|------|
|  |  | RMSE | RMSE | RMSE | RMSE |
|  |  | W/m² | W/m² | W/m² | W/m² |
| **Reference** | Norman et al., 1995 | **46,5** | **34,8** | **55,3** | **53,7** |
| **Net Radiation** | Anderson, 1997 | -- | **28,9** | **54,7** | **40,1** |
|  | Kustas and Norman, 1999 | 61,9 | 34,2 | 59,0 | 50,8 |
| **Conduction flux G** | Santanello et al., 2003 | -- | 36,4 | 62,6 | 75,1 |
|  | Chavez et al., 2005 | -- | 38,2 | **52,9** | 64,8 |
| **Transpiration** | Colaizzi et al., 2014 | -- | 99,1 | 74,5 | -- |
| **Surface resistance Rs** | Taconet et al., 1986 | -- | 44,7 | 71,6 | 43,0 |
| **Aerodynamic resistance Rah** | Taconet et al., 1986 | -- | 40,0 | **54,5** | -- |

**Fig. 1.**

| | | MAPD [%] | | | |
|---|---|---|---|---|---|
| | | H | | LE | |
| | | PT | PM | PT | PM |
| **Time resolution** | *Half-hourly data* | *21,2* | *69,8* | *26,7* | *33,0* |
| | *Daily average* | *16,7* | *66,2* | *18,5* | *26,8* |
| **Stage of growth** | *Rising* | *84,4* | *94,0* | *54,4* | *63,2* |
| | *Growth* | *18,0* | *86,4* | *29,4* | *41,4* |
| | *Max veg.* | *20,6* | *67,9* | *22,9* | *27,7* |
| | *Senescence* | *23,2* | *68,8* | *37,1* | *68,9* |
| | *Stress* | *24,7* | *59,4* | *32,5* | *35,2* |
| **Crop** | *Wheat* | *22,2* | *50,8* | *24,4* | *32,5* |
| | *Corn* | *19,1* | *79,9* | *29,9* | *31,1* |
| | *Sunflower* | *21,9* | *107,8* | *27,1* | *38,4* |

**Fig. 2.**

---

## Author Response (AR2)

**Answers to referee #2**

1) I am a bit surprised still that the simplified exponential extinction expression for the net radiation partitioning performed better than the separate longwave and shortwave partitioning developed in TSEB. Was this because the measured net radiation was used in the exponential extinction formulation (Eq. 9) instead of deriving net radiation from estimated net shortwave and estimated net longwave components? How net radiation was determined when applying Eq. (9) needs to be clarified in the text.

*Thanks for your comment, indeed this result is a bit surprising and this question arises. However we didn't use the measured net radiation but the derived net radiation from estimated shortwave and longwave components. This aspect was clarified in the text by adding the following text (p.4, l.25, after eq.9) :*
*"where the factor κ is set to 0.45 for spherical distribution of leaves following Roos (1991), and Rn is estimated from measured shortwave and longwave components".*

2) One other point to alert the authors concerning the poor performance of TSEB under sparse/bare soil conditions. They should cite the recent paper incorporating an improved soil resistance model/algorithm in TSEB for sparse canopies, which should be adopted in future TSEB analyses… Li et al 2018. Evaluating soil resistance formulations in thermal-based two source energy balance (TSEB) model: Implications for heterogeneous semiarid and arid regions Water Resources Research. https://doi.org/10.1029/2018WR022981.

*Thank you for this reference, it is actually relevant to the discussion. We figured out that the reference to Kustas et al., 2016 was also missing, and both were added in the text at section 3.1, p.13, l.10 :*
*"The soil resistance rs also plays an important role on bare and sparsely vegetated surfaces, and recent studies (Li et al., 2019; Kustas et al., 2016) showed that adapted formulation or modeling improved TSEB performances in arid or semi-arid conditions".*